# Potent acridone antimalarial against all three life stages of *Plasmodium*

Antimalarial therapeutics ideally should target all three major *Plasmodium* life cycle stages. Here we present an acridone antimalarial chemotype that is potent against blood, liver, and mosquito stages of malaria parasites. Attributes of lead candidate T111 include potent in vitro activity against cultured parasites, ex vivo activity against clinical isolates, oral single-dose cure in an asexual blood-stage rodent model, inhibition of sexual blood-stage parasites, activity against relapsing parasites in non-human primate liver cells, prevention of parasite development in mosquitoes, and synergy in combination with tafenoquine against blood- and liver-stage parasites. Analysis of parasites selected for resistance to T111 suggests inhibition of the mitochondrial electron transport chain, with a mechanism distinct from that of other antimalarials in use or under development. The safety profile of T111, including toxicology evaluations in rats, absence of hemolytic toxicity, and low genotoxicity and cardiotoxicity potential, demonstrates a favorable therapeutic index. Overall, T111 emerges as a promising candidate for treatment and prevention of malaria, with potential for single-dose cure of bloodstream infections, radical cure of liver infections, and interruption of transmission to mosquitoes.

Malaria has plagued humans for millennia and remains one of the deadliest diseases in modern times, resulting in approximately a quarter billion clinical cases and over a half million deaths annually[1]. The most vulnerable group is children under the age of five, with the estimation that a child dies from malaria roughly every minute. *Plasmodium falciparum* is responsible for the majority of malaria mortality worldwide, with *Plasmodium vivax* the most widespread human malaria parasite, and other plasmodial species that infect humans each presenting different challenges[2]. Despite commendable progress in past decades, malaria control continues to face many challenges, including alarming resistance to artemisinin (ART) and some artemisinin-based combination therapy (ACT) partner drugs[3,4], declining effectiveness of insecticides[5], safety concerns associated with the limited radical cure options for relapsing *P. vivax and Plasmodium ovale* infections[6,7], lack of transmission-inhibiting drugs[8], and inability to provide full immunity with newly implemented vaccines[9]. Ideally, a chemotherapy approach to combat all stages and species of

*Plasmodium* parasites would offer broad benefits toward the control and ultimate elimination of malaria[10,11].

Malaria is a mosquito-borne infectious disease caused by *Plasmodium* parasites, with life cycles involving both vertebrate hosts and mosquito vectors[11–13]. Human infection begins when an infected female *Anopheles* mosquito bites a person, injecting sporozoites which enter the blood stream and invade liver cells, multiply asexually, and mature into schizonts. Thousands of merozoites are then released into the blood, where they invade red blood cells, multiply asexually, and cause clinical illness. Some blood-stage parasites develop into sexual-stage parasites (called gametocytes) that are taken up by biting mosquitoes. Inside the mosquito, the parasites undergo sexual reproduction to develop into oocysts and then sporozoites. When the mosquito bites another human, sporozoites are injected, enabling continued transmission. Relapsing malaria, seen only with *P. vivax and P. ovale*, follows the reactivation of liver hypnozoites after extended periods of dormancy, leading to recurrent asexual bloodstream infection[14].

✉ e-mail: papiredd@pdx.edu; kellyja@ohsu.edu

Most of the antimalarial drugs currently in use target blood-stage parasites, which are responsible for clinical manifestations[11,15]. The only available causal prophylactics that target liver-stage infection are Malarone (atovaquone (ATV) + proguanil (PG)), primaquine (PQ), and recently approved tafenoquine (TQ). Of these, PQ and TQ are the only drugs that offer radical cures, killing hypnozoites to prevent relapsing *P. vivax* and *P. ovale* infections. In addition, low dose PQ is now recommended as a malaria transmission-inhibiting drug in some settings[7,16,17]. Unfortunately, PQ and TQ are both 8-aminoquinolines that cause hemolytic toxicity in patients with glucose-6-phosphate dehydrogenase (G6PD) deficiency, a common enzyme abnormality[18,19].

The advantages of antimalarial agents that eliminate malaria parasites in multiple life-cycle stages are self-evident, however the discovery of such highly desirable drugs remains an arduous challenge[11,20]. We have discovered and developed an acridone anti-malarial chemotype, with T111 as the lead candidate, that exhibits potent activity against liver (including hypnozoite), blood (both asexual and sexual), and mosquito stage parasites, with single-dose cure in a murine model as well as the potential to prevent relapsing infection and interrupt transmission.

## Results

### In vitro activity against asexual blood-stage *P. falciparum*
In our previous report[21], T111 exhibited potent in vitro antiplasmodial activity against a panel of asexual blood-stage *P. falciparum* parasites that included pan-sensitive strain D6, multi-drug resistant (MDR) strain Dd2, and ATV-resistant strain Tm90-C2B, with sub- to low-nanomolar $IC_{50}$ values. Considering the rising spread of ART resistance[22], here we tested T111, alongside dihydroartemisinin (DHA), against two recently isolated *P. falciparum* K13 mutant parasites, using a ring-stage survival assay (RSA), the Growth, Resistance, and Recovery Assay (GRRA)[23] to generate fine-scale dose-response curves ($GRRA_{50}$). SP045 and SP060 are K13 mutant clones from Rwanda[24,25]. SP045 (V555A) is relatively susceptible to ART (ART-S), and SP060 (R561H) is highly ART-resistant (ART-R)[25]. As demonstrated in Fig. 1A and B, ART-R SP060 has a DHA $GRRA_{50}$ value of 25 nM and ART-S SP045 has a $GRRA_{50}$ of 7.7 nM. T111 exhibits approximately 10- to 29-fold greater potency than DHA, with sub-nanomolar effectiveness against both parasites, and indicates no cross-resistance between T111 and DHA in these clinically relevant isolates ($GRRA_{50}$ = 0.78 nM for SP045 and 0.87 nM for SP060). T111 was also tested against the ART-R lab-derived *P. falciparum* clones DR1 and DR4, which were selected in the Dd2 strain with escalating DHA pressure[26]. DHA had decreased activity against these two resistant parasites ($IC_{50}$ = 90 nM for DR1 and 173 nM for DR4 compared to 8.3 nM for Dd2)[26]. As shown in Fig. 1C, T111 displayed excellent potency, against Dd2 ($IC_{50}$ = 0.028 nM), DR1 ($IC_{50}$ = 0.11 nM), and DR4 ($IC_{50}$ = 0.11 nM).

### Ex vivo activity against asexual blood-stage *P. falciparum* clinical isolates
The activity of T111 against fresh *P. falciparum* isolates collected from patients diagnosed with *falciparum* malaria were evaluated in Burkina Faso and Uganda[27,28]. T111 demonstrated picomolar activity, with median $IC_{50}$ values of 0.062 nM (0.0013–0.49 nM, $n$ = 78) in Burkina Faso and 0.049 nM (0.0038–0.66 nM, $n$ = 81) in Uganda, with ex vivo activities similar to in vitro activities against the Dd2 and 3D7 control strains (Fig. 1D). Ex vivo testing in the field provides real-time data on drug potency. Importantly, all patient isolates were highly susceptible to T111, with sub-nanomolar $IC_{50}$ values.

### In vivo oral efficacy against asexual blood-stage *P. yoelii* in a rodent model
To investigate oral efficacy against blood-stage infections, we employed a well-established model[21,29–31] using 4–5-week-old CF1 mice inoculated with *P. yoelii* ($3.5 \times 10^4$ per inoculum via tail vein injection).

As shown in Supplementary Fig. 1, T111 (via oral treatment) exhibited excellent antimalarial efficacy with low $ED_{50}$ and $ED_{90}$ values in the 4-day model ($ED_{50}$ = 0.35 and $ED_{90}$ = 0.50 mg/kg/day in female mice; $ED_{50}$ = 0.33 and $ED_{90}$ = 0.46 mg/kg/day in male mice), and was curative at 10 mg/kg/day for four consecutive days. With interest in a single-dose curative regimen, we also tested T111 administered orally once, 24 h post-inoculation[31–33]. One of four mice treated with a single oral dose of 40 mg/kg, and all of four mice treated with 50 mg/kg were cured, with no parasites detected over 28 days (Fig. 1E).

### In vitro activity against sexual blood-stage *P. falciparum* gametocytes
The activity of T111 against sexual blood-stage gametocytes was investigated after asexual stage *P. falciparum* NF54 parasites were stressed by starvation to induce gametocytogenesis[34]. The drug was added daily on Days 5 to 7 post gametocyte induction when the gametocytes were approximately stage II to III, and gametocytemia (stage V gametocytes) was determined on Day 14, when control parasites had matured into late-stage gametocytes. T111 inhibited the formation of stage V gametocytes by 75% and 93% at 1 μM and 10 μM, respectively (Fig. 1F).

### Standard membrane feeding assay (SMFA)
The transmission-inhibitory activity of T111 was investigated in a standard direct membrane feeding assay[35], in which the drug was included during mosquito feeding with blood samples from a naturally infected *P. falciparum* NF54 strain gametocyte carrier. T111 inhibited oocyst development in a dose-dependent manner and eliminated oocyst formation at 4.0 ng/mL (9.5 nM) concentration (Fig. 1G).

### Tarsal contact assay
In a tarsal contact assay[36], female *Anopheles gambiae* G3 strain mosquitoes were allowed to land on a thin film surface incorporated with a T111 prodrug at 5% by weight (structure shown in Supplementary Fig. 4) for 6 min and 1 h later fed a blood meal carrying *P. falciparum* NF54 gametocytes. Midguts of the infected mosquitoes were dissected on Day 7. There was a significant, nearly complete reduction in oocyst prevalence (92.5% inhibition) and intensity (97.6% inhibition) in mosquitoes after brief tarsal contact with the T111 prodrug, whereas nearly all control mosquitoes showed a high intensity infection (Fig. 1H).

### In vitro activity against relapsing *Plasmodium cynomolgi* in rhesus hepatocytes
*P. cynomolgi* is a non-human primate (NHP) malaria parasite closely related to *P. vivax* and is widely used as surrogate model for relapsing malaria due to its ability to form dormant liver-stage hypnozoites. To evaluate the activity of T111 against relapsing liver-stage malaria, we employed a high-throughput in vitro assay using *P. cynomolgi* sporozoites infecting primary rhesus hepatocytes[37], assessing both prophylaxis (prevention of liver-stage establishment) and radical cure (elimination of existing hypnozoites and developing liver-stage schizonts) dosing paradigms. Under prophylactic conditions, T111 exhibited potent activity against the establishment of liver-stage hypnozoites, comparable or superior to that of the reference compounds TQ[38], ATV[39], and the *Plasmodium* PI4 kinase (PI4K) inhibitor KDU691[40,41] (Table 1). In addition, prophylactic activity inhibiting schizont formation with low-nanomolar $IC_{50}$ values was observed. Considering radical cure treatment against dormant hypnozoites, the potency of T111 was superior to the known hypnozoitocidal reference drug TQ for inhibition of both hypnozoites and schizonts. Cytotoxicity of T111 was assessed in parallel by photometric measurements of host hepatocyte viability, revealing moderate in vitro profile against rhesus hepatocytes ($CC_{50}$ = 0.56 μM; Table 1); however, the cytotoxicity $IC_{50}$ value vs human HepG2 hepatocyte cells was > 200 μM (Supplementary Table 1).

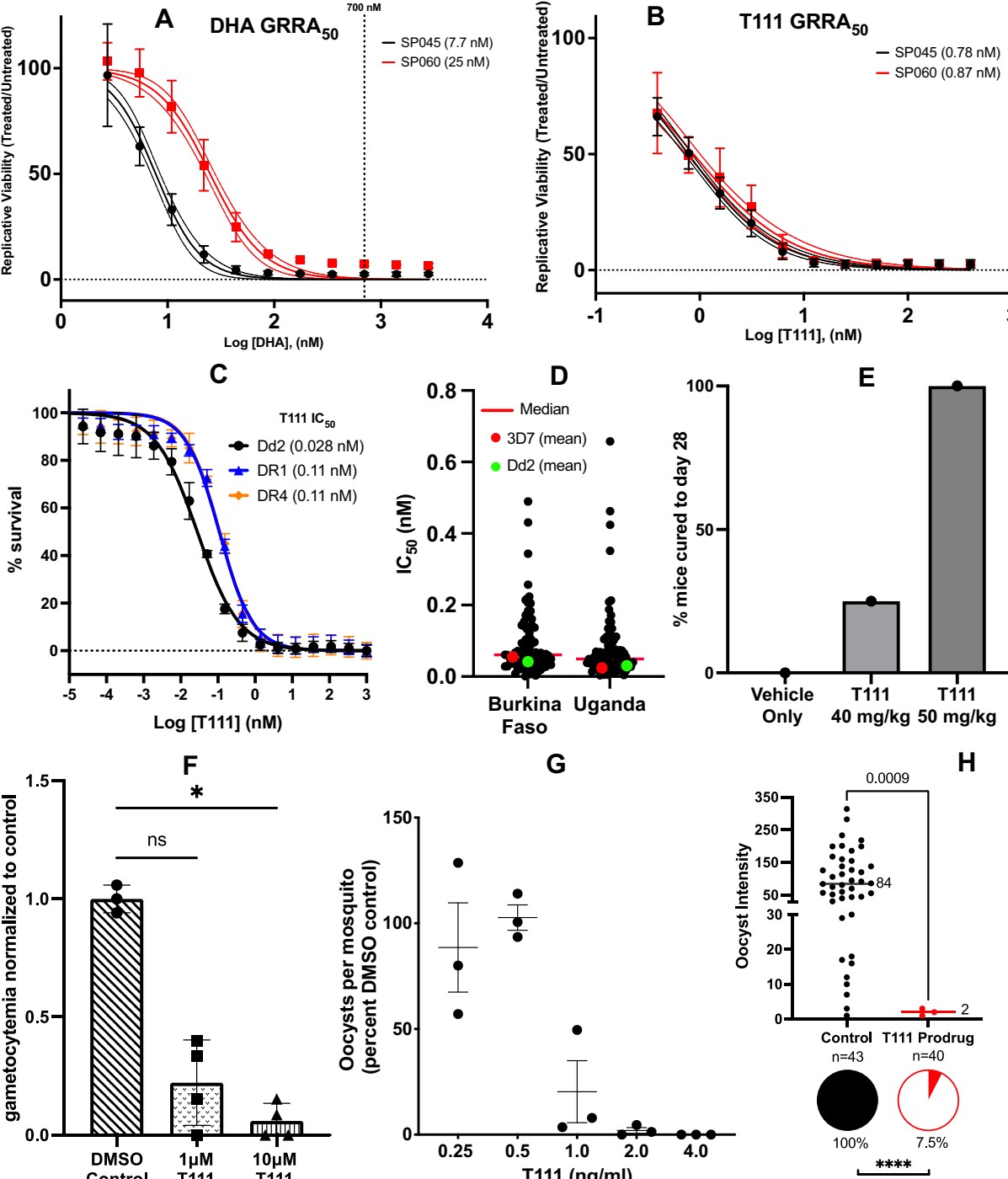

**Fig. 1 | Antimalarial activities of T111 against blood- and mosquito-stage *Plasmodium* parasites. A, B** GRRA dose–response curves for DHA and T111 against *P. falciparum* isolates SP060 (ART-resistant) and SP045 (ART-sensitive). Center values: mean; error bars: SD. Results from $n = 3$ independent biological replicates, each in technical duplicate. Dashed lines: 95% CI; vertical dashed line in (**A**): 700 nM concentration used in the traditional RSA. **C** In vitro potency of T111 against laboratory-adapted ART-resistant *P. falciparum*. Center values: mean; error bars: SD. Results from $n = 3$ independent biological replicates, each in technical triplicate. **D** Ex vivo susceptibilities of *P. falciparum* clinical isolates. Dots represent individual isolates ($n = 78$ Burkina Faso; $n = 81$ Uganda). Control strain (3D7 and Dd2) values represent the means of replicate assays. Replicates were not performed for clinical isolates due to limited resources and only one blood sample being available per patient. **E** In vivo efficacy of oral T111 against blood-stage *P. yoelii* in 4–5-week-old female CF1 mice. Bars: percentage of mice cured. Results from $n = 4$ independent biological replicates per group. **F** In vitro

inhibition of sexual blood-stage *P. falciparum* gametocytes. Data: mean ± SD of $n = 2$ independent biological replicates, each in technical duplicate. Statistical significance was determined by a Kruskal-Wallis with Dunn's: ns ($p = 0.1612$), * ($p = 0.0163$). Biological replication was limited ($n = 2$) due to the specialized nature of the assay. **G** In vivo inhibition of *P. falciparum* oocyst development via direct membrane feeding. Center values: mean; error bars: SEM. Dots ($n = 3$ independent biological replicates) represent mean oocyst counts from 12–23 mosquitoes per trial. Statistical significance was determined by a Friedman test ($p = 0.011$). **H** In vivo inhibition of *P. falciparum* oocyst development via tarsal contact. Pie charts: infection prevalence; statistical significance was determined by a two-tailed Fisher's exact test ($p < 0.0001$). Intensity graph: oocysts per infected mosquito; center lines: median; statistical significance was determined by a two-tailed Mann-Whitney test ($p = 0.0009$). Results from $n = 2$ independent biological replicates (infection trials) with 83 mosquitoes total (control $n = 43$; T111 $n = 40$).

**Table 1 | In vitro antiplasmodial activity against liver-stage *P. cynomolgi* in primary rhesus hepatocytes**

| | inhibition IC$_{50}$ (µM)[a] vs NHP *P. cynomolgi* | | | | cytotoxicity CC$_{50}$ (µM)[a,d] vs rhesus hepatocytes |
| | prophylactic mode[b] | | radical cure mode[c] | | |
| drug | hypnozoite | schizont | hypnozoite | schizont | |
|---|---|---|---|---|---|
| T111 | 0.093 (1.50) | 0.018 (3.65) | 0.50 (1.26) | 0.28 (1.46) | 0.56 (1.29) |
| TQ | 0.18 (2.48) | 0.17 (2.18) | 8.1 (1.74) | 9.8 (1.00) | >20 |
| ATV | >1.2 | 0.0093 (1.19) | 2.46 (1.12) | 2.25 (1.17) | 3.96 (1.39) |
| KDU691 | 0.12 (1.40) | 0.098 (1.68) | >20 | 0.19 (1.67) | >20 |
| MAD | <0.00098 | <0.00098 | 0.021 (1.53) | 0.041 (1.59) | 0.36 (2.15) |

*TQ*, tafenoquine, *ATV* atovaquone, *MAD* maduramicin.

[a]IC$_{50}$ and CC$_{50}$ values represent the geometric mean (x/÷ geometric SD) of n ≥ 3 independent biological replicates, each performed in technical duplicate.
[b]Prophylactic mode: drugs were administered 1 hour post sporozoite-infection on Day 0 and continued until Day 2.
[c]Radical cure mode: drugs were administered daily from Day 4 through Day 7 post-sporozoite infection.
[d]Cytotoxicity vs rhesus hepatocytes was determined in the radical cure mode.

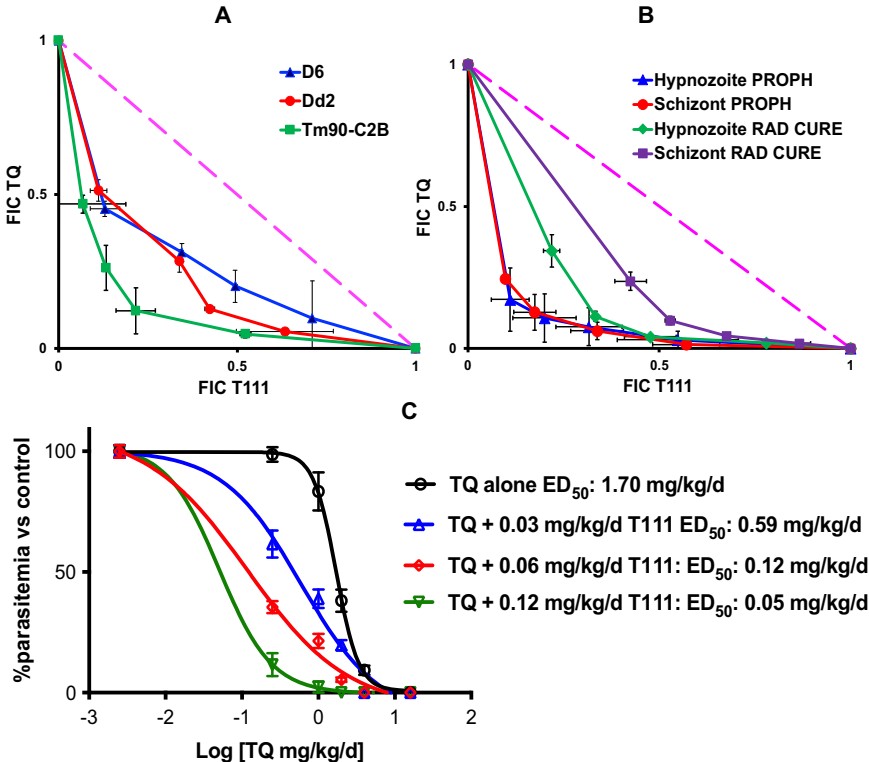

**Fig. 2 | Drug-drug interactions between T111 and TQ against *Plasmodium* parasites. A** In vitro isobologram analysis of the interaction between T111 and TQ against asexual blood-stage *P. falciparum*. FIC values are presented as the mean, and error bars represent the SEM. Results are derived from *n* = 2 independent biological replicates, each performed in technical quadruplicate. The dashed line represents the theoretical additive effect (ΣFIC = 1.0). **B** In vitro isobologram analysis of the interaction between T111 and TQ against liver-stage rhesus *P. cynomolgi*. FIC values are presented for prophylactic (PROPH) and radical cure (RAD CURE) treatment modes against hypnozoites and schizonts. Center values represent the mean, and error bars represent the SEM. Results are derived from *n* = 3 independent biological replicates, each performed in technical quadruplicate. The dashed line represents the theoretical additive effect (ΣFIC = 1.0). **C** In vivo potentiation of TQ oral efficacy by T111. Efficacy was evaluated against *P. yoelii* infection in a blood-stage 4-day suppression murine model, using four- to five-week-old female CF1 mice. Center values represent the mean, and error bars represent the SEM. Results are derived from *n* = 4 independent biological replicates per treatment group.

## In vitro synergy of the T111/TQ combination against asexual blood-stage *P. falciparum* parasites

Synergy was assessed by isobolar analysis with a well-established fixed ratio platform[42,43], using a SYBR Green based assay[44]. Drug interactions were quantified by calculating the Fractional Inhibitory Concentration (FIC) for each drug, defined as the IC$_{50}$ of the drug in combination divided by its IC$_{50}$ alone. The ΣFIC index (the sum of the individual FICs) was used to determine the nature of the interaction, where a ΣFIC value < 1.0 indicates synergy. As shown in Fig. 2A, relative synergy between T111 and TQ was observed against pan-sensitive D6 (ΣFIC

index = 0.69), MDR Dd2 (ΣFIC index = 0.62), and ATV-resistant Tm90-C2B (ΣFIC index = 0.46).

## In vitro synergy of the T111/TQ combination against relapsing *P. cynomolgi* parasites

Using the in vitro *P. cynomolgi* assay described above[37], for prophylaxis, synergy was evident in the inhibition of both hypnozoite and schizont development, with ΣFIC index values for the TQ/T111 combination at 0.39 and 0.41, respectively. For radical cure, synergy was also observed in inhibiting hypnozoite and schizont

**Table 2 | In vivo single-dose cure efficacy of the T111/TQ combination against *P. yoelii* in mice**

| drug combinations[a] (mg/kg, single dose) | | parasitemia burden on Day 5[b] | number of mice cured on Day 28 (% survival)[c] |
|---|---|---|---|
| TQ | T111 | | |
| 15 | 0 | 0 | 0/4 (0%) |
| 20 | 0 | 0 | 1/4 (25%) |
| 30 | 0 | 0 | 3/4 (75%) |
| 40 | 0 | 0 | 4/4 (100%) |
| 0 | 20 | 0 | 0/4 (0%) |
| 0 | 30 | 0 | 0/4 (0%) |
| 0 | 40 | 0 | 1/4 (25%) |
| 0 | 50 | 0 | 4/4 (100%) |
| 5.0 | 5.0 | 0 | 0/4 (0%) |
| 5.0 | 10 | 0 | 0/4 (0%) |
| 5.0 | 15 | 0 | 3/4 (75%) |
| 5.0 | 20 | 0 | 4/4 (100%) |
| 10 | 5.0 | 0 | 0/4 (0%) |
| 10 | 10 | 0 | 3/4 (75%) |
| 10 | 15 | 0 | 4/4 (100%) |
| 10 | 20 | 0 | 4/4 (100%) |
| 15 | 5.0 | 0 | 4/4 (100%) |
| 15 | 10 | 0 | 4/4 (100%) |
| 15 | 15 | 0 | 4/4 (100%) |
| 15 | 20 | 0 | 4/4 (100%) |
| 20 | 5.0 | 0 | 4/4 (100%) |
| 20 | 10 | 0 | 4/4 (100%) |
| 20 | 15 | 0 | 4/4 (100%) |
| 20 | 20 | 0 | 4/4 (100%) |
| Vehicle only group | | 38% | 0/4 (0%) |

[a] 4–5-week-old female CF1 mice (*n* = 4 per group) were inoculated with $3.5 \times 10^4$ *P. yoelii* parasitized red blood cells. Drugs were administered as a single oral dose (*p.o.*) 24 h post-infection.
[b] Parasitemia burden was assessed via Giemsa-stained thin blood smears; "0" indicates no detectable parasites at the limit of detection.
[c] Mice were monitored for 28 days; "Cured" is defined as the absence of patent parasitemia through Day 28. Values in parentheses represent the survival rate of mice through Day 28.

development, with ΣFIC index values at 0.58 and 0.72, respectively (Fig. 2B).

### In vivo potentiation of TQ against blood-stage *P. yoelii* in the 4-day suppression murine model

We compared the activity of TQ alone and in the presence of sub-therapeutic oral dosages of T111 in the 4-day suppression model. T111 potentiated the oral efficacy of TQ, with an ~34-fold reduction in $ED_{50}$ values at 0.12 mg/kg/day (Fig. 2C). The dose-response curves for oral efficacy of T111 alone are shown in Supplementary Fig. 1.

### In vivo single-dose cure synergy of the T111/TQ combination against *P. yoelii* in mice

We also investigated the combination of T111 and TQ as a single-dose cure in the *P. yoelii* model. TQ achieved single-dose cure in all treated mice at 40 mg/kg (Table 2). T111 achieved single-dose cure in all treated mice at 50 mg/kg. Various lower dosage combinations of TQ with T111 achieved single-dose cures (Table 2).

### In vivo synergy of the T111/TQ combination against liver-stage *Plasmodium berghei* in mice

We previously reported in vivo oral efficacy of T111 against liver-stage *Plasmodium* infection[21]. A real-time in vivo imaging system (IVIS) using transgenic bioluminescent parasites to quantify parasite development was employed to study liver-stage infection in live

anesthetized *P. berghei* infected mice[45–47]. Potential synergy between T111 and TQ was investigated using the same model. Luciferase-expressing *P. berghei* sporozoites were inoculated in 4–6-week-old Albino C57BL/6 mice on Day 0, with oral doses of drugs administered on Day −1 (1 day prior to inoculation), Day 0 (day of inoculation), and Day 1 (1-day post-inoculation) and bioluminescence signals were measured at 24 and 48 h after inoculation for liver-stage development and at 72 h for blood-stage infection in anesthetized mice. Strong bioluminescence signals were detected in untreated mice at both 24 and 48 h in the liver, followed by intense whole-body signals at 72 h (Fig. 3). Treatment with TQ at 1 mg/kg/day or 2.5 mg/kg/day, and with T111 at 1 mg/kg/day or 4 mg/kg/day did not clear liver-stage infection, and all treated mice developed blood-stage infection. Treatment with TQ or T111 at 10 mg/kg/day provided full protection from liver-stage infections. Treatment with the combination of 1 mg/kg/day TQ + 4 mg/kg/day T111 provided full protection, without visible parasites in the liver at 24 and 48 h or the blood over 31 days after inoculation. The same results were observed with the combination of 2.5 mg/kg/day TQ + 1 mg/kg/day T111.

### In vitro metabolic stability and in vivo pharmacokinetic (PK) profile

T111 was evaluated for in vitro metabolic stability by measuring the disappearance of the parent compound after incubation with pooled human, mouse, and rhesus liver microsomes (HLM, MLM, and RLM)[48,49]. T111 demonstrated good metabolic stability in all microsomal systems ($t_{1/2} > 60$ min in HLM and MLM; and $t_{1/2} = 34.0$ min in RLM). In vivo PK studies of T111 were conducted following a single oral administration in both male and female ICR-CD1 mice at 40 mg/kg with blood and liver samples taken at 11 time points, using well-established methods[21,31,50]. PK profiles of T111 were similar in male and female mice. Considering the reported PK data for ATV ($t_{1/2}$ in mice plasma ~13 h[51]), T111 demonstrated a relatively long half-life in both liver and plasma, with higher concentration in liver than plasma (Table 3), which is consistent with its excellent in vivo oral efficacy against liver-stage infection[21]. The development of prodrugs of T111 with enhanced PK profiles is underway.

### In vitro safety evaluations

Assessment of cytotoxicity against human hepatic HepG2 cells using the MTT assay[52] showed no apparent toxicity, with $IC_{50} > 200$ μM, indicating a selectivity index > 10,000 (Supplementary Table 1). hERG (human-ether-a-go-go-related gene) channel inhibition assessment using an automatic parallel patch clamp system[53] showed low risk of cardiotoxicity, with estimated $IC_{50}$ value higher than 100 μM (Supplementary Table 2). The Ames test against *Salmonella typhimurium* TA98 and TA100 strains with and without S9 activation[54] showed negative results at the tested concentration of 10 μM. No increase above the background reversion rate was observed, indicating a low risk of mutagenicity (Supplementary Fig. 2). Evaluation of eryptosis as a marker for potential drug induced hemolytic toxicity, characterized by cell shrinkage, oxidative stress, and phosphatidylserine translocation to cell surface[55], did not show the toxicity seen with TQ against G6PD-deficient RBCs (Fig. 4).

### In vivo toxicology and toxicokinetic (TK) analysis in rats

Murine blood-stage and liver-stage efficacy studies included observations of animal weight, grooming, posture, and locomotion. No overt clinical toxicity or behavior change was observed in any T111 treated mice (highest tested dose: 30 mg/kg × 4 days and 80 mg/kg × 1 day). In addition, safety evaluation of T111 was conducted in male and female rats, with both single dose and repeat dose toxicology studies. In the maximum tolerated dose (MTD) determination study, a single oral dose of T111 was administered by gavage at 30, 100, 200, and 400 mg/kg to male and female Sprague Dawley rats, and animals were observed

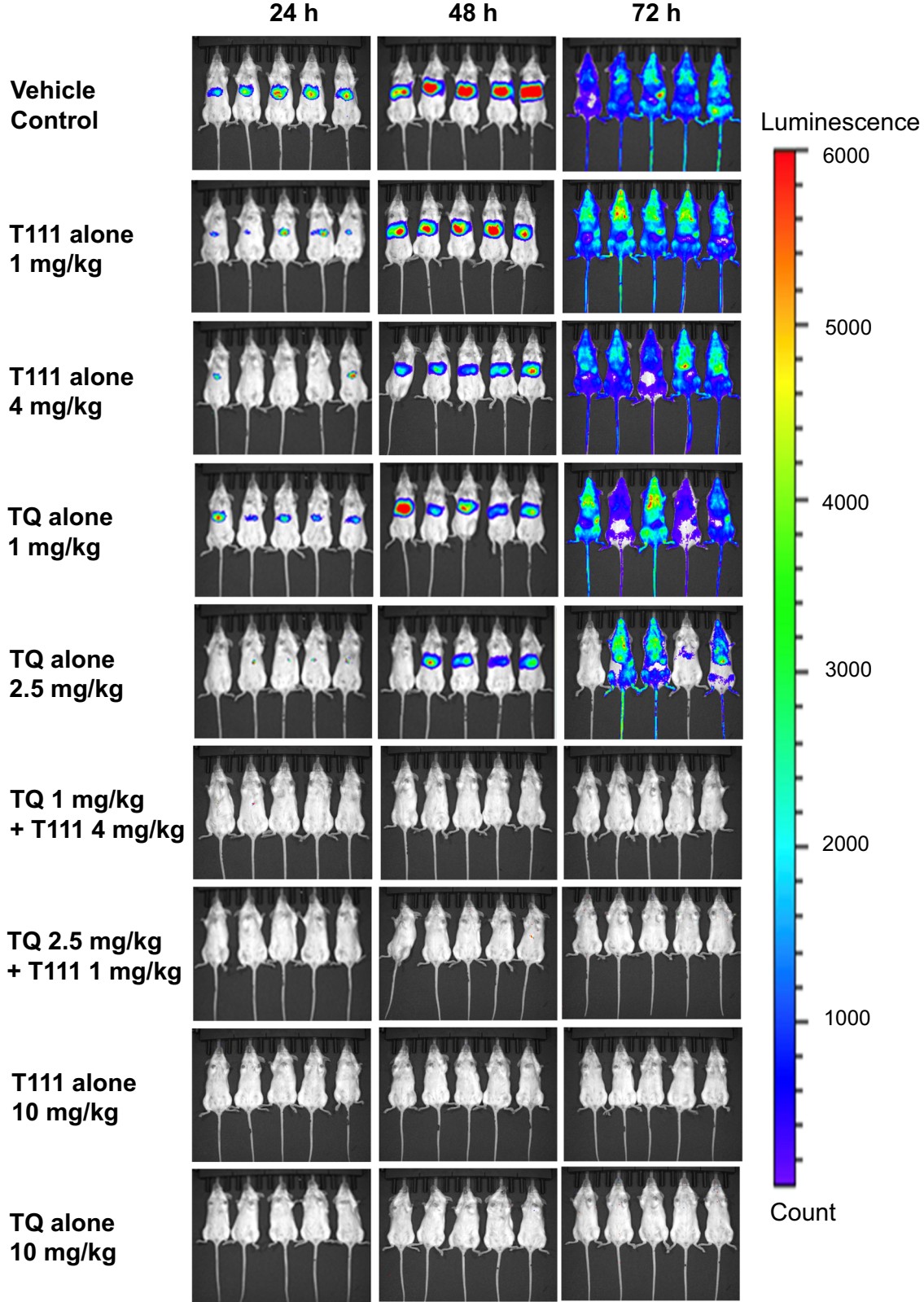

**Fig. 3 | Bioluminescence and real-time in vivo imaging of parasite load.** Biolu-minescence imaging of 4- to 6-week-old female Albino C57BL/6 mice ($n = 5$ per group) following inoculation and treatment with T111, TQ, or the T111/TQ combination. PEG-400 was used as a vehicle control. Intensity of biolumi-nescence is indicated by the right-hand-side color scale (photons/sec).

for two days. Clinical observations were performed immediately, 2–4 h, and 24 h post dosing. T111 was well tolerated, with no clinical signs suggesting toxicity. A single dose MTD could not be established due to limited solubility of T111 in the vehicle (PEG-400), but it is considered to be greater than 400 mg/kg, the highest dose tested. Based on the observed effects and T111 solubility limitations in the MTD studies, the dose levels selected for 7-day repeat dose studies were 25, 100, and 400 mg/kg. Male and female Sprague Dawley rats were given T111 daily

**Table 3 | Key PK parameters of T111 in plasma and liver following a single oral dose of 40 mg/kg administration in both male and female mice[a]**

| mice | matrix | $C_{max}$ (ng/mL) | $T_{max}$ (h) | $AUC_{last}$ (ng.h/mL) | $AUC_{inf}$ (ng.h/mL) | $t_{1/2}$ (h) | CL/F (mL/h/kg) |
|---|---|---|---|---|---|---|---|
| male | plasma | 6.8 | 4.0 | 156 | 180 | 26.5 | 202,794 |
| | liver | 1704 | 1.0 | 25,969 | 26,044 | 8.30 | 1408 |
| female | plasma | 11.3 | 4.0 | 169 | 185 | 17.5 | 216,187 |
| | liver | 1503 | 8.0 | 23,384 | 23,495 | 10.5 | 1702 |

[a]PK parameters were obtained using noncompartmental analysis via the Phoenix WinNonlin software package; $n = 3$ male and $n = 3$ female ICR-CD1 mice (4–5 weeks old) were used per time point. $C_{max}$ maximum plasma or hepatic concentration, $T_{max}$ time to $C_{max}$, $t_{1/2}$ apparent elimination half-life, $AUC_{last}$ area under the concentration–time curve from 0 up to the last sampling time at which a quantifiable concentration is found, $AUC_{inf}$ area under the concentration–time curve from 0 up to infinity, CL/F apparent oral clearance.

for 7 consecutive days via oral gavage, then euthanized and evaluated for body weight changes, clinical signs, plasma drug concentrations, organ weights, gross and microscopic examinations of selected tissues, and clinical pathology (hematology, clinical chemistry). These repeated daily oral administrations were well tolerated, albeit with minor clinical signs, specifically, shoveling and ruffled fur in T111-treated animals. Repetitive motions were observed in T111-treated females at 100 mg/kg and 400 mg/kg, and in one control male and one treated male in the 400 mg/kg treatment group. Pink discharge in the eye and/or nostril areas (chromodacryorrhea) was observed with increased frequency in T111-treated females relative to controls, as well as two males from the 25 mg/kg treatment group and one male from the 100 mg/kg treatment group. No significant alterations in body weight, organ weight, clinical pathology, or histopathology were observed. For TK analysis, T111 plasma levels were determined at 6 time points on Day 1 and Day 7 in male and female rats. T111 was not completely cleared within 24 h, indicating potential accumulation upon repeated daily dosing (Supplementary Table 3). Maximum systemic exposure was obtained in the 100 mg/kg dose group, likely due to the limited solubility at 400 mg/kg, which was administered as a suspension (the lower doses were administered as solutions). The TK profile of T111 in these studies was consistent with the PK parameters in mice (Table 3), which indicated a long half-life and rapid absorption[21].

**Drug resistance selection**

To gain insight into the mechanism of action and propensity for drug resistance selection, we cultured *P. falciparum* Dd2 strain under continuous and incrementally increasing concentrations of T111. Over 16–18 months, the parasites acquired resistance to T111 in an ordered and sequential manner (Fig. 5). Because earlier iterations of development of our acridone chemotype had a presumed target of *Plasmodium* cytochrome *b* (cyt *b*)[30], we directly sequenced *pfcytb* in parasites emerging from drug selections with decreased susceptibilities to T111. Multiple mutations in the cyt *b* gene were identified in these resistant parasites (Fig. 5). Initial selection led to modest resistance in parasites with a single mutation, V259L or G131S. Further selection yielded parasites with the combination of these two mutations, and subsequent selection with high nanomolar concentrations of T111 yielded parasites with 3 or 4 *pfcytb* mutations, including two novel mutations V140I and I119L. All mutations selected, except L87V, are located in the cyt *b* $Q_o$ site[56].

To confirm that cyt *b* mutant haplotypes confer T111 resistance, we tested susceptibility of the mutant lines to T111 and other compounds known to target the mitochondrial electron transport chain (ETC) (Table 4). T111 retained low nanomolar potency against parasites with two cyt *b* mutations, but a third mutation conferred high level resistance. T111-resistant mutant lines did not show significant cross resistance with ATV ($Q_o$ site inhibitor), but had various degrees of cross resistance with other $Q_o$ site inhibitors (ELQ-400 and myxothiazol (MYX)). Two T111-resistant mutant lines, Dd2-$C_0^{I119L-G131S-V259L}$ and Dd2-$C_1^{G131S-V140I-V259L}$, exhibited significant cross resistance with ELQ-300 ($Q_i$ site inhibitor) and DSM265 (DHODH inhibitor), and sequencing revealed that they also carried the *pfdhodh* C276F mutation. There was no cross resistance with the

clinically deployed antimalarials chloroquine (CQ), piperaquine (PIP), lumefantrine (LUM), or DHA (Supplementary Table 4), which do not act against the ETC.

**Whole genome sequencing (WGS) and analysis**

To determine whether T111 selection induces mutations in other genomic regions and to assess their relationship with the T111 resistance phenotype, six clones carrying multiple *pfcytb* mutations after prolonged selection with T111 and the unselected parental strain were evaluated with WGS (Supplementary Table 5). All mutations previously identified by direct sequencing of *pfcytb* and *pfdhodh* were confirmed by WGS. The only gene in which mutations were observed in all independently selected drug resistance lines was *pfcytb*, offering strong evidence that *Pf*CYTB is a primary target of T111[11,57]. We observed other mutations elsewhere in the genome, but not in each selected line. Additional mutations occurred in two clones in the *pfdhodh* gene, consistent with findings from Sanger sequencing. Importantly, *Pf*DHODH was the only protein with selected mutations other than CYT B that is a known drug target. Mutations in *pfdhodh* have previously been associated with resistance to cyt *b* inhibitors[58]. Supplementary Table 5 identifies 15 other genes that acquired mutations in at least one of the drug selected lines. These sporadic mutations were likely attributable to spontaneous genetic variation that developed over a year of continuous in vitro culture and not to selection by T111.

**Cross-resistance profile with *Plasmodium* mitochondrial electron transport chain (ETC) inhibitors with different site preferences**

To study cross-resistance between T111 and other ETC inhibitors, we used a panel of *P. falciparum* parasites with different cyt *b* $Q_o$/$Q_i$ site sequences (Table 5). Clinical isolate Tm90-C2B carries a point mutation at the $Q_o$ site of cyt *b* (Y268S) that confers ATV resistance[59]. The Dd2-$D_1^{I22L}$ clone was selected by culturing *P. falciparum* Dd2 under ELQ-300 drug pressure, leading to an I22L mutation in the $Q_i$ region of cyt *b*, contributing to loss of sensitivity to ELQ-300[60]. *P. falciparum* transgenic D10yDHODH parasites, which bypass the parasite's ETC fueled pyrimidine synthesis pathway, are therefore completely resistant to all mitochondrial ETC inhibitors[61,62]. We also included pan-sensitive (D6) and MDR (Dd2) strains of *P. falciparum* as reference parasites. T111 exhibited decreased activity against ATV-resistant parasites, but retained nanomolar potency ($IC_{50}$ vs Tm90-C2B = 4.0 nM). Interestingly, T111 had enhanced activity against parasites resistant to the $Q_i$ site inhibitor ELQ-300 ($IC_{50}$ of T111 vs Dd2-$D_1^{I22L}$ = 0.00073 nM). In contrast, ATV and ELQ-300 had decreased activity against Tm90-C2B (ATV $IC_{50}$ = 8538 nM) and Dd2-$D_1^{I22L}$ (ELQ-300 $IC_{50}$ = 245 nM), respectively. Similar to the ETC inhibitors ATV, ELQ-300, and DSM1[61,63], T111 had reduced activity against D10yDHODH, with a resistance index >1000 (vs D6 or Dd2), confirming that T111 targets the mitochondrial ETC.

To further validate the target, T111 was tested in the 4-day suppression rodent model using an ATV-resistant *P. yoelii* clone (ATV$^r$) harboring Y268S mutations corresponding to the $Q_o$ site of cyt $bc_1$ corresponding to known ATV-resistant mutations in *P. falciparum*[64] and

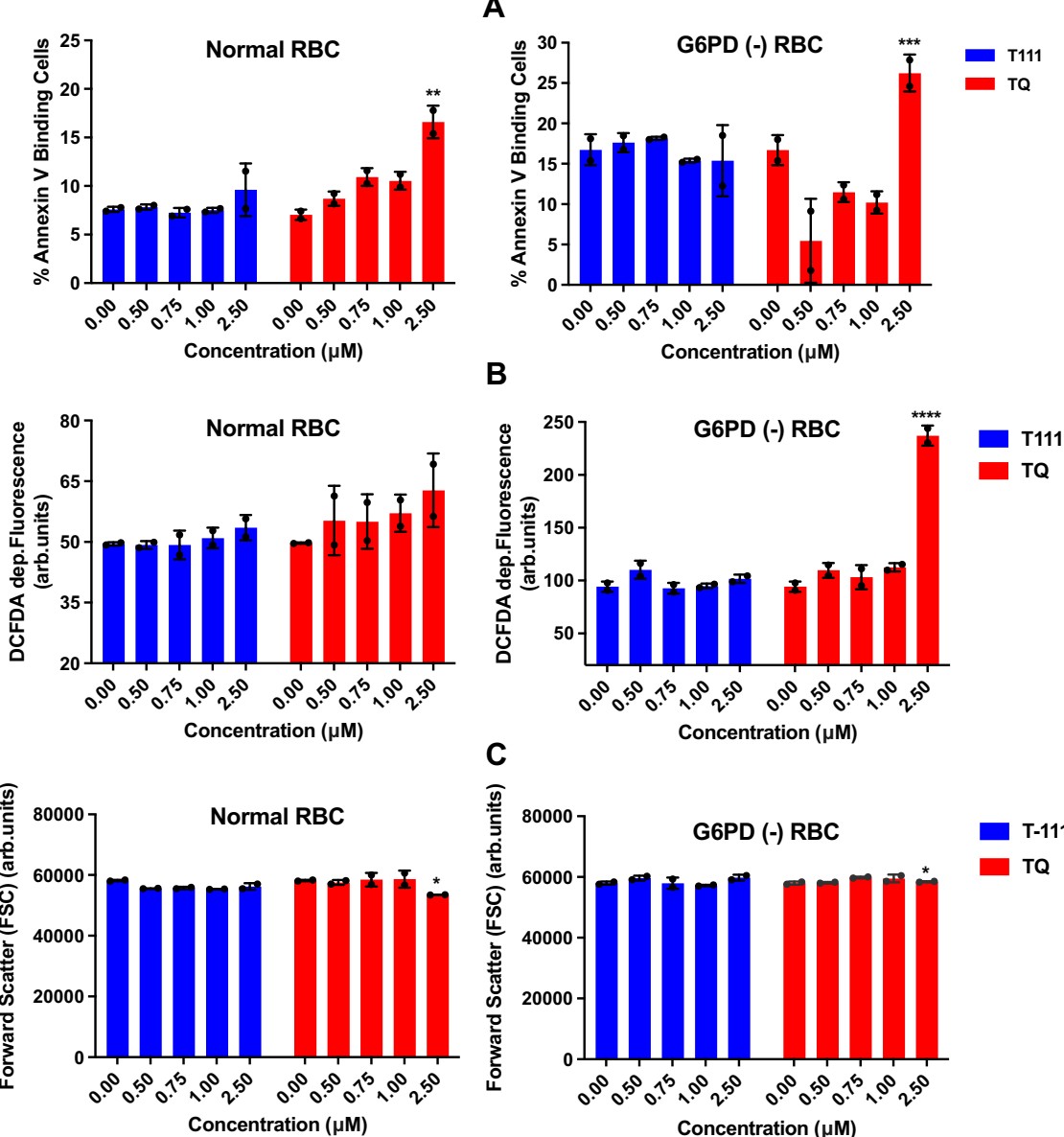

**Fig. 4 | Effect of T111 and TQ on eryptosis. A** Phosphatidylserine exposure shown as red blood cell (RBC) annexin-V-binding, **B** reactive oxygen species (ROS) formation shown as 2′,7′-dichlorodihydrofluorescein diacetate (DCFDA) fluorescence, and **C** RBC size/volume shown as forward scatter (FSC) following 48 h incubation with drugs. Data are presented as mean ± SD ($n = 1$ biological replicate; performed in technical duplicate) with individual data points overlaid to show data distribution. Biological replicates were not performed due to limited resources and the restricted availability of primary human erythrocyte samples from volunteer donors. Statistical significance was determined by two-way ANOVA. $p$ values represent the interaction between drug treatment and concentration: for (**A**), $^{**}p = 0.008$ for normal RBC and $^{***}p = 0.0008$ for G6PD (−) RBC; for (**B**), $^{****}p < 0.0001$ for G6PD (−) RBC; and for panel C, $^{*}p = 0.03$ for both normal and G6PD (−) RBC.

an ELQ-300 resistant *P. yoelii* clone (ELQ-300$^r$) containing an I22L mutation at the $Q_i$ region of *cyt bc$_1$*, corresponding to known ELQ-300 resistant mutations in *P. falciparum* generated in vitro[64]. As shown in Supplementary Table 6, although both ATV$^r$ and ELQ-300$^r$ *P. yoelii* strains were highly resistant to their respective parent compounds, ATV$^r$ parasites remained fully sensitive to ELQ-300 and vice versa[64]. T111 exhibited slightly reduced oral efficacy against ATV-resistant parasites ($ED_{50} = 0.24$ mg/kg/day vs wild type; $ED_{50} = 0.31$ vs ATV$^r_{Y268S}$), with no cross-resistance with ELQ-300 ($ED_{50} = 0.25$ vs ELQ-300$^r_{I22L}$), which is consistent with our observations from in vitro evaluation.

### Docking analysis of T111-selected *cyt b* mutants
We used AutoDock 4.0 to model the predicted binding of T111 to *P. falciparum cyt b*. Based on the homologous crystal structure of *S. cerevisiae cyt b*, the $Q_o$ binding site in *P. falciparum cyt b* was modeled by substituting amino acids at corresponding positions[65]. T111 readily docked into the modeled *P. falciparum cyt b* $Q_o$ binding site, where residues G131 and V259 surrounded the ligand (Supplementary Fig. 3). However, introduction of mutations identified in T111-selected resistant strains revealed potential steric clashes: the V140I substitution clashed with the head group of T111, whereas the I119L substitution clashed with the tail end of the compound, destabilizing its binding (Supplementary Fig. 3). Consistent with these predictions, parasite lines harboring either mutation exhibited the highest T111 IC$_{50}$ values (Table 4). Together, the IC$_{50}$ measurements and molecular docking results indicate that selection of the third mutation confers high-level resistance to T111.

### Discussion
Major findings have emerged since the initial discovery of the acridone antimalarial chemotype[21,30], leading to the development of a preclinical

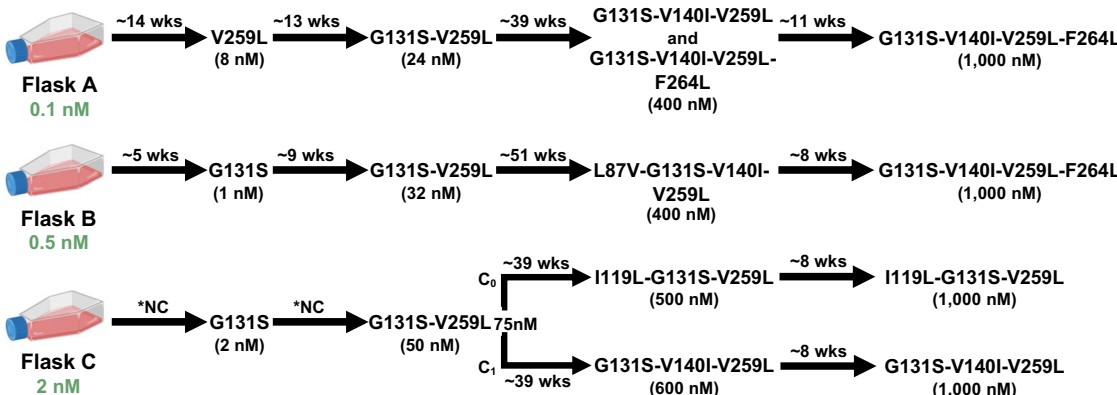

**Fig. 5 | Selection of *cyt b* mutations in *P. falciparum* cultures treated with T111.** Schematic representation of the stepwise acquisition of mutations in the *cytochrome b* gene (*cyt b*) across three independent selection flasks (Flasks A, B, and C). Initial T111 starting concentrations are indicated in green below each flask. Black arrows indicate the progression of selections, with the approximate duration (wks) and the T111 concentration at the time of parasite cloning (in parentheses) noted for each step. Specific amino acid substitutions are identified by their position in the cyt b protein. *NC, not calculated; wks, weeks.

### Table 4 | Antiplasmodial activity of ETC inhibitors against *P. falciparum* mutant T111 lines

| parasite line | inhibition IC$_{50}$ vs *P. falciparum* (nM)[a] | | | | | |
|---|---|---|---|---|---|---|
| | **T111** | **MYX** | **ATV** | **ELQ-400** | **ELQ-300** | **DSM265** |
| Dd2 | 0.15 ± 0.023 | 24 ± 3.4 | 0.44 ± 0.052 | 4.0 ± 0.45 | 38 ± 5.0 | 3.2 ± 0.30 |
| Dd2[131S] | 0.78 ± 0.081 | 383 ± 38 | 0.34 ± 0.049 | 4.2 ± 0.41 | 20 ± 5.8 | 1.9 ± 0.22 |
| Dd2[259L] | 4.4 ± 0.71 | 64 ± 13 | 6.6 ± 1.9 | 24 ± 6.7 | 23 ± 4.3 | 3.3 ± 1.0 |
| Dd2[131S-259L] | 12 ± 2.5 | 975 ± 139 | 1.5 ± 0.17 | 28 ± 13 | 26 ± 5.2 | 2.8 ± 1.2 |
| Dd2-A[131S-140I-259L] | 74 ± 5.1 | 192 ± 11 | 4.7 ± 0.44 | 133 ± 17 | 52 ± 5.5 | 3.2 ± 0.52 |
| Dd2-A[131S-140I-259L-264L] | 98 ± 12 | 16 ± 4.1 | 15 ± 2.0 | 80 ± 15 | 35 ± 3.2 | 1.2 ± 0.075 |
| Dd2-B[87V-131S-140I-259L] | 129 ± 6.3 | 373 ± 11 | 10 ± 1.4 | 177 ± 3.1 | 67 ± 4.6 | 3.5 ± 0.37 |
| Dd2-B[131S-140I-259L-264L] | 114 ± 13 | 34 ± 3.3 | 11 ± 1.8 | 126 ± 32 | 25 ± 2.7 | 2.8 ± 0.30 |
| Dd2-C$_0$[119L-131S-259Lb] | 144 ± 12 | 1538 ± 110 | 15 ± 0.46 | 255 ± 62 | 221 ± 50 | 795 ± 235 |
| Dd2-C$_1$[131S-140I-259Lb] | 223 ± 36 | 723 ± 141 | 25 ± 7.2 | 501 ± 101 | 225 ± 73 | 563 ± 203 |

*MYX* myxothiazol, *ATV* atovaquone.
[a]IC$_{50}$ values represent the mean ± SEM of n ≥ 3 independent biological replicates.
[b]Parasite lines harboring a mutation in *pfdhodh*.

### Table 5 | Cross-resistance profile of T111 against *Plasmodium* parasites with diverse ETC resistance genotypes/phenotypes

| drug | in vitro IC$_{50}$ (nM)[a] vs *P. falciparum* | | | | |
|---|---|---|---|---|---|
| | **D6** | **Dd2** | **Tm90-C2B** | **Dd2-D$_1$[I22L]** | **D10yDHODH** |
| T111 | 0.028 ± 0.00036 | 0.042 ± 0.0042 | 4.0 ± 0.076 | 0.00073 ± 0.000088 | >2500 |
| ATV | 0.073 ± 0.010 | 0.13 ± 0.012 | 8538 ± 65 | 0.0070 ± 0.0025 | >2500 |
| ELQ-300 | 3.5 ± 0.23 | 3.1 ± 0.15 | 1.3 ± 0.084 | 245 ± 27 | >2500 |
| CQ | 14 ± 0.70 | 166 ± 5.8 | 211 ± 7.6 | 209 ± 26 | 79 ± 16 |

*ATV* atovaquone, *CQ* chloroquine.
[a]IC$_{50}$ values represent the mean ± SEM of *n* = 3 independent experiments, each performed in quadruplicate.

candidate (T111) with the following attributes: 1) in vitro inhibition of *P. falciparum* asexual blood-stage growth at picomolar concentrations with a favorable selectivity index[21]; 2) ex vivo picomolar activity against fresh isolates from *P. falciparum* infected African patients; 3) in vitro inhibition of sexual blood-stage *P. falciparum* gametocyte formation;

4) in vivo oral curative efficacy (including single-dose cure) in an erythrocytic *P. yoelii* murine model; 5) in vitro prevention of *P. berghei* development in human hepatocytes at low nanomolar concentrations[21]; 6) in vitro prevention and inhibition of relapsing *P. cynomolgi* hypnozoite and schizont development in rhesus

hepatocytes at nanomolar concentrations; 7) in vivo protection from *P. berghei* liver infection and cure of blood infection via oral administration in mice[21]; 8) in vitro synergy with TQ against erythrocytic *P. falciparum*; 9) in vitro synergy with TQ against relapsing *P. cynomolgi* in rhesus hepatocytes; 10) in vivo potentiation of TQ oral efficacy against blood-stage *P. yoelii* murine malaria; 11) in vivo synergy with TQ against liver-stage *P. berghei* infection in mice; 12) in vivo inhibition of *P. berghei* and *P. falciparum* sporozoite formation in mosquitoes[66]; 13) in vivo prevention of oocyst formation in mosquitoes at nanomolar concentrations; 14) in vivo disruption of oocyst formation in mosquitoes by tarsal contact; 15) promising safety profiles, with low in vitro hERG inhibition, negative Ames results, and no observed T111 induced eryptosis in both normal and G6PD deficient red blood cells; 16) good tolerability in animals without observed toxicity; 17) feasible DMPK profiles, with rapid absorption, long half-life, and high drug concentrations in the liver[21]; and 18) high potency against both laboratory-adopted and clinical ART-resistant *P. falciparum* strains. Furthermore, an efficient four-step synthesis from commercially available starting materials enables cost-effective large-scale production of T111.

Safety profiling of T111 indicated lack of hemolytic toxicity or toxicological findings in experimental animals. We demonstrated potency of T111 against liver-stage *Plasmodium* hypnozoites and strong synergy with TQ in preventing liver-stage infection (including hypnozoites) and treating blood-stage infection (including single-dose cure) using multiple in vitro and animal model assays. These findings suggest that the addition of T111 allows dose reduction for TQ, which might mitigate G6PD liabilities. Given the paucity of hypnozoitocidal candidates and narrow safety profiles with TQ, the radical cure activity of T111 is promising, whether on its own or as a synergistic partner to improve the TQ therapeutic window for treatment and prevention of *vivax* malaria, including in G6PD deficient patients. Further testing of the anti-relapse efficacy of T111 and its potential synergy with TQ in an NHP model is underway.

Previously, the Goodman group reported that T111 completely inhibited sporozoite development when drugs were fed to infected mosquitoes, with potency against both *P. berghei* and *P. falciparum* superior to that of ATV and all other tested drugs (either in clinical use or under development)[66]. Here, we report that T111 completely prevents *P. falciparum* oocyst formation in a direct membrane feeding assay that simulates a blood meal taken from a treated human and disrupts oocyst formation in a tarsal contact assay that mimics uptake of the drug after landing on an antimalarial infused bed net. These findings offer direct evidence of transmission-inhibiting potential for T111, establishing a potential role for this antimalarial chemotype.

The *Plasmodium* mitochondrial ETC, including the cytochrome $bc_1$ complex and DHODH, is essential for parasite survival in human and mosquito stages, and a validated drug target[67]. The Q cycle inside the $cyt\ bc_1$ complex has two quinone-binding sites, the quinol oxidation site ($Q_o$) and quinone reduction site ($Q_i$), located on opposite sides of the membrane and linked by a transmembrane electron-transfer chain. ATV is the only antimalarial ETC inhibitor in clinical use, and it targets the $Q_o$ site, with resistance associated with point mutations at this site[59]. Selection with T111 yielded resistance associated with the sequential acquisition of mutations in the $Q_o$ site. However, there are important distinctions between T111 and ATV, both genetically and functionally. First, T111 is highly refractory to resistance selection as parasites required 16–18 months of constant selection pressure to become highly resistant to T111, whereas ATV resistance can be easily generated in a few weeks[68,69]. Second, T111 selection did not generate the clinically important Y268S mutation that renders ATV inactive. T111 retained nanomolar potency against parasites with two *cyt b* mutations, and the selection of a third mutation was necessary to confer high level resistance. Third, T111 remained highly potent against ATV-resistant parasites, and reciprocally, T111-resistant parasites did not show significant cross resistance with ATV. Unlike T111, ATV is

inactive against *Plasmodium* liver-stage relapsing hypnozoites[70] and sexual blood-stage gametocytes[71]. Extensive cross-resistance profiling using parasites with various resistance profiles and ETC inhibitors with different $Q_o/Q_i$ site preferences demonstrated that T111 had picomolar to nanomolar potency against parasites resistant to other $Q_o$ or $Q_i$ site inhibitors. These mechanistic investigations suggest a complex mode of action for T111. In vitro selected T111 resistance was enhanced by a mutation in *pfdhodh*, suggesting that T111 interactions extend beyond the $Q_o$ site of the ETC. Targeting subtle interactions within the ETC that are crucial to antimalarial potency and resistance has allowed us to overcome ATV resistance, adding support for the ETC as an important target for new antimalarials. Development of new ETC inhibitors is further encouraged by the inability of ATV-resistant parasites to survive in mosquitoes, inhibiting the spread of resistance[35,72].

We have developed an antimalarial chemotype that has potent activity against the main life cycle stages of *Plasmodium* parasites, with our lead candidate (T111) meeting all five current target candidate profiles (TCP) proposed by Medicines for Malaria Venture (MMV): molecules that clear asexual blood-stage parasitemia (TCP-1); are active against hypnozoites (TCP-3); are active against hepatic schizonts (TCP-4); that interrupt transmission by targeting parasite gametocytes (TCP-5); and that interrupt transmission by targeting the insect vector (TCP-6)[11,16]. In addition, T111 exhibits synergistic interaction with TQ against both liver- and blood-stage parasites, as well as optimal safety profiles and DMPK properties, with a suggested mechanism of action distinct from that of other antimalarials currently in use or under development. The combination of these features in T111 is unique and promising, potentially offering a drug with distinctive properties for antimalarial treatment and chemoprevention.

Given the overall profile, including killing relapsing liver-stage parasites and a lack of apparent hemolytic toxicity, the acridone chemotype represented by the T111 prototype has the potential to fill a crucial gap in the current antimalarial pipeline. However, limitations with the T111 prototype, such as suboptimal solubility and moderate oral efficacy, necessitate further optimization. We have initiated the development of T111 prodrugs, using a validated approach[73], and demonstrated improved PK profiles and enhanced oral efficacy in preliminary studies. Validation of in vivo anti-relapse efficacy and synergy with TQ in *P. cynomolgi*-infected NHPs for the selected acridone prodrug is underway. The results, as well as other in-depth preclinical evaluations (e.g., PK/PD predictions, PK studies in malaria-infected mice, and drug uptake and efflux kinetics), will be reported when available.

## Methods
The research was conducted under an IACUC-approved animal use protocol in an AAALAC International-accredited facility with a Public Health Services Animal Welfare Assurance and in compliance with the Animal Welfare Act and other federal statutes and regulations relating to laboratory animals. In vivo asexual blood-stage antimalarial studies were carried out at VA Portland Health Care System (VAPORHCS), and the procedures were conducted under protocols approved by the VAPORHCS's IACUC. In vivo liver-stage antimalarial studies and in vivo PK were carried out at Walter Reed Army Institute of Research (WRAIR), and the procedures were conducted under protocols approved by the WRAIR/Naval Medical Research Command (NMRC)'s IACUC in accordance with national and Department of Defense guidelines. In vivo safety and TK studies were carried out at SRI International (SRI), and the procedures were conducted under protocols approved by the SRI's IACUC.

### Synthesis of T111 and its prodrug
The original synthetic method for preparing T111 was described in our previous publication[21]. Details of the revised synthetic approach (Supplementary Fig. 4), including the experimental procedures, and

structural characterization data (Supplementary Methods and Supplementary Figs. 5–14) for the scale-up production of T111, as well as the synthetic method for its prodrug, are provided in the Supplementary Information.

### In vitro GRRA against ART-resistant clinical isolates

T111 was evaluated against two *P. falciparum* K13 mutants (SP045 and SP060) originally obtained from symptomatic malaria patients in the Huye District, Southern Province, Rwanda, during a 2019 molecular surveillance study. These isolates carry the validated R561H Kelch 13 propeller domain mutation associated with delayed parasite clearance in East Africa[24,25]. Using the 6-hour GRRA[23], all strains were thawed, frozen, and cultured under standard conditions at 5% hematocrit. Samples parasitemia was measured by flow cytometry. Parasite cultures that were predominantly segmented schizonts were synchronized using a single Percoll layer method[23]. Cultures that met quality control requirements of ≥70% early ring stage and ≥1% parasitemia were adjusted to 0.25% parasitemia at 2% hematocrit. Treated samples were exposed starting at 2800 nM to DHA in 0.02% DMSO and 10 subsequent two-fold dilutions along with a no-drug control, or starting at 400 nM of T111 in 0.02% DMSO followed by 10 two-fold dilutions; untreated (no-drug control) samples were exposed to the same amount of DMSO (0.02%) alone. The $GRRA_{50}$ implements 11 RSA experiments to generate a dose-response curve for each parasite line per drug. Master mixes of samples were aliquoted into 2 technical replicates of 275 μL each in 96 well plates (2 treated each condition, 2 untreated). At 6 h post-exposure, 200 μL of media was removed from the wells and replaced with 200 μL of incomplete media and spun for 3 min at 800 g to rinse. The rinse was repeated a total of 3 times, after which 200 μL complete media was added back to the wells. At 120 h post-treatment, thin microscopy smears were made using 2 μL of culture for each sample and 20 μL of culture was transferred to PCR strips and stored at −80 °C until qPCR quantification. A standard curve of known parasitemia was generated using the NF54 parasite line for each qPCR quantification for conversion from Cycle threshold ($C_t$) to parasitemia and comparison between qPCR experiments. Replicate viability was calculated according to the GRRA method[23]. Each individual reaction mixture and cycling conditions were set to previously used methods. $C_t$ was calculated using AB1 SDA 2.4.1. software. All graphs and regressions were performed in the software GraphPad Prism 10 (version 10.5.0). We find that commonly reported thresholds (cutoffs) for resistance (e.g., 1% survival in traditional RSA) may not be broadly applicable, especially with lower thresholds subject to potential artifacts due to markedly low survival rates of ART-sensitive and low-level ART-resistant parasites. Regular RSA relies on a single drug concentration, 700 nM. To gain a more fine-scale quantification to compare drug susceptibilities with relevance to therapeutic success and failure, we developed the $GRRA_{50}$ to generate full dose-response curves for precise quantitative comparisons.

### Ethical statement for Rwandan clinical isolates

The *P. falciparum* clinical isolates used in this study (including SP045 and SP060) were originally collected during a molecular surveillance study conducted between September and December 2019 in the Huye District, Southern Province, Rwanda. Ethical Clearance for the primary collection and subsequent use of these isolates was granted by the Rwanda National Ethics Committee (Kigali, Rwanda, approval ref no. 416/RNEC/2017 and 686/RNEC/2019). Prior to enrollment, written informed consent was obtained from all adult participants or from the parents or legal guardians of child participants. Additionally, written assent was obtained from all minor participants aged 7–18 years, in accordance with the guidance of the Rwanda National Ethics Committee. All samples were de-identified prior to use in the current study.

### In vitro drug susceptibility assay against ART-resistant parasites

Drug susceptibility assay against ART-resistant *P. falciparum* clones was performed using a SYBR Green-based assay[44,74]. Synchronized ring stage parasites were cultured for 72 h in the presence of the test drug. The assay was performed in 96-well plates at 2% hematocrit and 0.5% starting parasitemia. Each experiment was conducted in triplicate and repeated with three biological replicates. Parasite growth at 72 h was quantified via SYBR Green I staining. The obtained counts were plotted against the logarithm of the compound concentration, and a curve was fitted with non-linear regression (sigmoidal dose-response/variable slope equation) to determine $IC_{50}$ (half-maximal inhibitory concentration) values. The data was analyzed using GraphPad Prism software 8.0.

### Ex vivo drug susceptibility assay against clinical isolates

The ex vivo antiplasmodial activity of T111 was tested against fresh *P. falciparum* isolates using a 72-h growth inhibition assay with SYBR Green detection[27,28]. Isolates were collected from patients with uncomplicated *falciparum* malaria in Bobo-Dioulasso, Burkina Faso and Tororo, Uganda. For parasite culture, parasitemia was identified with Giemsa-stained thin films using a light microscope. Only samples with at least 0.3% parasitemia were analyzed. Isolates were stored at 4 °C and assayed within 24 h of collection. To measure ex vivo drug susceptibilities, a 72 h microplate growth inhibition assay was used with SYBR Green detection. Study compounds were dissolved in DMSO as 10 mM stocks and stored at −20 °C. Drugs were serially diluted, including drug-free and parasite-free control wells, with concentrations optimized to capture full dose-response curves. Cultures were diluted with uninfected $O^+$ erythrocytes to achieve 0.2% parasitemia and 2% hematocrit. After 72 h, wells were resuspended, and culture from each well was transferred to 96-well plates containing SYBR Green lysis buffer and mixed. Plates were incubated for 1 h in the dark at room temperature, and fluorescence was measured with a plate reader (485 nm excitation and 530 nm emission). $IC_{50}$ values were derived by plotting fluorescence intensity against log drug concentration and fit to a non-linear curve using a four-parameter Hill equation in Prism (version 9.0).

### Ethical statement for Burkina Faso and Uganda clinical isolates

The *P. falciparum* clinical isolates used for ex vivo susceptibility testing in this study were obtained from symptomatic patients with uncomplicated malaria in Burkina Faso and Uganda. Ethical clearance for the collection and subsequent use of these isolates was granted by the Institutional Ethics Committee of the Institut de Recherche en Sciences de la Santé (IRSS) (Bobo-Dioulasso, Burkina Faso), the Makerere University School of Biomedical Sciences Research Ethics Committee (Kampala, Uganda), and the Uganda National Council for Science and Technology. In Burkina Faso, isolates were collected during the 2021 and 2022 malaria transmission seasons from patients seeking treatment at health centers in Bobo-Dioulasso. In Uganda, isolates were collected longitudinally from government health facilities in the Tororo, Busia, Mbale, and Agago districts (2010–2024). Prior to enrollment, written informed consent was obtained from all adult participants or from the parents or legal guardians of child participants. Written assent was also obtained from all minor participants (aged 8–17 years) in accordance with the guidance of the respective National Ethics Committees. All samples were de-identified and assigned unique study identifiers prior to laboratory analysis.

### In vivo antimalarial efficacy against blood-stage *P. yoelii* in rodent model

The efficacy of T111 and the T111/TQ combination was determined using two well-established murine models against *P. yoelii* parasites: 1) a 4-day suppression model[21,29–31]; and 2) a single-dose cure model[31–33]. Animals were housed at the VAPORHCS in Thoren Maxi-Miser

microisolator cages (78 sq. in. floor space) with a maximum density of five females or four males per cage. Cages were maintained on a 12-h light/dark cycle and provided with autoclaved nesting materials (nestlets) for environmental enrichment. All mice had ad libitum access to food and water. Four- to five-week-old female and male outbred CF1 mice (Charles River Laboratories, Raleigh, NC, USA) were randomly assigned to treatment and control groups using a simple randomization procedure (e.g., computer-generated random numbers) to ensure balanced distribution of baseline body weights and starting parasitemia levels across all cohorts. Groups of four mice ($n = 4$, mean body weight of each group was ~30 g) were infected intravenously with $3.5 \times 10^4$ *P. yoelii* (Kenya strain, MRA-428) from a donor animal. Drug administration commenced the day after the animals were inoculated (Day 1). The test drug or the drug combination was dissolved in PEG-400 and administered by oral gavage once daily for 4 successive days in the 4-day suppression model, but only once on Day 1 in the single-dose cure model. Blood was collected via the tail vein with the aid of a syringe needle on Day 5 and then at weekly intervals through Day 28. Blood films were Giemsa-stained and examined microscopically to determine the level of parasitemia. All mice were observed daily to assess clinical signs. Animals with observable parasitemia following the experiments were euthanized; animals cleared of parasitemia from their bloodstream were observed daily with assessment of parasitemia performed weekly until Day 28, at which point, the animal(s) were scored as cured of the infection and then humanely euthanized. All treated mice with a negative smear on Day 28 were considered cured (100% protection). $ED_{50}$ and $ED_{90}$ values (mg/kg/day) were derived graphically from the dose required to reduce the parasite burden by 50% and 90%, respectively, relative to drug-free controls.

### Animals and ethical statement at VAPORHCS

In vivo asexual blood-stage antimalarial efficacy studies were conducted at the VAPORHCS under a protocol reviewed and approved by the VA Portland Institutional Animal Care and Use Committee (IACUC). This work was originally approved under local animal protocol #6067-22 (IRBNet ID #1673475-5) on March 26, 2022. A triennial renewal was subsequently granted under protocol #6067-24 (IRBNet ID #1673475-13), effective December 18, 2024, which remains valid through December 17, 2027. This research also received federal oversight and approval from the U.S. Army Medical Research and Development Command (USAMRDC) Animal Care and Use Review Office (ACURO) under protocol #PR210491.e001. The protocol was originally approved by ACURO on May 10, 2022, with the most recent renewal approval granted on February 26, 2025. All research was conducted in an AAALAC International-accredited facility in strict accordance with the Animal Welfare Act and the *Guide for the Care and Use of Laboratory Animals* (NRC Publication, Eighth edition)[75]. All research described herein with all relevant ethical regulations.

### In vitro antiplasmodial activity against sexual blood-stage *P. falciparum* gametocytes

*P. falciparum* NF54 was used to generate gametocytes. The erythrocytic stage of the parasite was cultured following standard laboratory protocols. Gametocyte production was carried out as described by Miura et al[34]. *P. falciparum* NF54 asexual parasites were cultured in RPMI 1640 medium (KD Medical) containing *L*-glutamine, 25 mM HEPES, and 50 μg/mL hypoxanthine, and supplemented with 0.21% sodium bicarbonate (from a 7.5% stock solution: KD Medical), 2 μg/mL gentamicin (from a 10 mg/mL stock solution: KD Medical), and 0.25% (w/v) Albumax II. Cultures were maintained in washed human O$^+$ red blood cells at a final hematocrit of 2%[76]. For gametocyte cultures, parasites were induced using serum-free human media[34]. Gametocyte cultures were maintained in washed human O$^+$ red blood cells at a final hematocrit of 5%. Cultures were initiated at an asexual parasitemia of

1.5% in serum-free gametocyte media and allowed to crash and form mature gametocytes over 14 days[77,78]. All cultures were maintained at 37 °C in a Billups chamber under a gas mixture of 90% $N_2$, 5% $CO_2$, and 5% $O_2$, with media changes daily. On Days 5 through 7 post gametocyte induction, gametocyte cultures were treated with T111 at concentrations of 1 μM or 10 μM. Fresh T111 was added at each media change. DMSO diluted in gametocyte media was used as the control. To remove T111, cultures were washed once with RPMI 1640, centrifuged at $1000 \times g$ for 3 min, and resuspended in warm serum-free gametocyte media without drug. Gametocytemia was assessed on Day 14 post-induction using blood smears stained with the Hema 3™ Manual Staining System (Fisher) and analyzed by light microscopy.

### Membrane feeding assay (SMFA) in mosquitoes

The SMFA assay was performed using *P. falciparum* NF54 gametocytes produced in vitro[35]. *P. falciparum* NF54 gametocytes were grown in human blood cultured in RPMI 1640 GlutaMax medium (Gibco) supplemented with 25 mM HEPES, hypoxanthine, and human serum[35]. De-identified human blood and serum were obtained from the Australian Red Cross Blood Service. Mature (stage V) gametocytes were quantified in Giemsa-stained blood smears. Immediately prior to infection, a feeding mixture comprising 45% human serum and 55% human blood containing 0.15% stage V gametocytes and the desired concentration of drug or vehicle control, was prepared and maintained at 37 °C. Adult female *Anopheles stephensi* (MR4) mosquitoes reared under the standard insectary conditions[79] were placed in mesh-topped cups and allowed to feed on the gametocyte mixtures in prewarmed glass feeders wrapped with parafilm. Mosquitoes that failed to feed in 20 min were removed. Mosquito midguts were dissected 7 days after infection, stained in 0.2% mercurochrome[35], and oocysts were counted using a light microscope.

### Ethical statement for human blood product used for SMFA assay

The use of de-identified human blood products for the in vitro SMFA assay was conducted at the University of Melbourne. This work was performed under the University of Melbourne Human Research Ethics Committee (HREC) project ID number 22013. All blood products were provided by the Australian Red Cross Lifeblood in accordance with national ethical guidelines and specific deeds of agreement.

### Tarsal contact assay in mosquitoes

The tarsal assay using a thin film was performed[36]. For preparation of a thin film, 25 mL xylenes and 1.0 g low density polyethylene polymer were combined on the lid of a glass petri dish (6 cm) and heated at 150 °C until the polymer was dissolved. Xylenes were occasionally replenished upon evaporation. A solution of xylenes containing T111 prodrug was then added to the above solution. Excess xylenes were evaporated slowly and the resulting crude polymer films were allowed to dry overnight. The crude polymer thin film was then put underneath the glass petri dish, and the assembled apparatus was heated on a hot plate with a weight placed on top. The resulting extruded polymer thin film containing the 5% (w/w) T111 prodrug in low-density polyethylene (LDPE) was allowed to cool and removed from the petri dish lid. Next, the T111 prodrug film was placed in a 6 cm glass petri dish fitted with a transparent lid to contain mosquitoes. 4–7-day old female *Anopheles gambiae* G3 strain mosquitoes were aspirated into this container and allowed to land on the T111 prodrug thin film for six min. After exposure, the mosquitoes were immediately transferred to a cage and one hour later provided with cultured *P. falciparum* NF54 gametocyte-infected erythrocytes using a membrane feeder. Infected mosquitoes were housed in a secure infection glovebox for the remainder of the experiments and provided with 10% w/v glucose water solution *ad libitum*. On Day 7 post-infection, mosquitoes were aspirated into 80% ethanol and incubated at −20 °C for 10 min before being transferred to 1X PBS for dissection. Midguts were dissected into iX PBS, stained with

0.2% w/v mercurochrome (mercury dibromofluorescein disodium salt, Sigma-Aldrich) in double-distilled water for 14 min, then mounted in 0.02% mercurochrome for microscopy. Midguts were imaged and oocysts counted at 100× magnification on an Olympus Inverted CKX41 microscope to quantify oocyst prevalence and intensity.

## In vitro liver-stage *P. cynomolgi* assay

The liver-stage *P. cynomolgi* assay was designed to identify anti-hypnozoite drugs utilizing a 384-well in vitro culture system[31,37]. Briefly, cryopreserved primary simian hepatocytes (donor lots XGB, OQB, and QHO: BioIVT, Inc.) were seeded in collagen-coated 384-well plates and infected with *P. cynomolgi bastianellii* (B strain) sporozoites dissected from infected *Anopheles dirus* mosquitoes. Test compounds were solubilized in 100% DMSO and serially diluted 2-fold to generate up to a 12-point dose-response curve, maintaining a final DMSO concentration of 0.1% across all wells. Two distinct treatment modes were evaluated. For the prophylactic assay, to assess the prevention of infection, compound exposure began 1-hour post-infection with daily media changes on Days 1 and 2. For the radical cure assay, to assess activity against established liver forms, drugs were administered daily from Day 4 through Day 7 post-infection. In both treatment modes, the highest concentration tested was 20 μM for KDU691, T111, and TQ, 10 μM for ATV and MAD. For readout on Day 8, plates were fixed and stained via immuno-fluorescence to visualize parasite forms (hypnozoites and schizonts) and hepatocyte nuclei, and images were acquired using an Operetta CLS high-content imaging system (PerkinElmer). Host hepatocyte viability was measured in parallel to determine compound toxicity. Parasite forms were automatically quantified using Harmony 4.9 software, and percent inhibition and toxicity were calculated relative to in-plate DMSO vehicle controls. $IC_{50}$ and $CC_{50}$ (half-maximal cytotoxic concentrations) were determined by fitting dose-response data to a four-parameter logistic (variable slope) model using a custom Python script incorporating a grid search algorithm.

## Ethical statement for *P. cynomolgi* sporozoites production at Armed Forces Research Institute of Medical Sciences (AFRIMS)

*Anopheles dirus* mosquitoes infected with the B strain of *P. cynomolgi* were produced by the Department of Entomology at the AFRIMS (Bangkok, Thailand). All procedures involving non-human primates (NHP) and mosquitoes were conducted under an animal use protocol reviewed and approved by the USAMD-AFRIMS Institutional Animal Care and Use Committee (IACUC) and received formal oversight and approval from the U.S. Army Medical Research and Development Command (USAMRDC) Animal CARE and Use Review Office (ACURO) on September 19, 2022. Animals were maintained in strict accordance with the *Guide for the Care and Use of Laboratory Animals* (NRC Publication, Eighth edition)[75] and the Animal for Scientific Purposes Act (Thailand). The USAMD-AFRIMS animal care and use program is fully accredited by AAALAC International. In compliance with U.S. Department of Defense and institutional security policies, specific internal protocol numbers are maintained by the USAMD-AFRIMS IACUC Office and are available for independent verification by the Editorial Office upon request.

## In vivo antimalarial efficacy against liver-stage *P. berghei* in rodent model

The liver-stage efficacy of T111 or the T111/TQ combinations was evaluated as prophylaxis in a murine model using the IVIS system[21,30,31,45,46]. Four- to six-week-old female inbred Albino C57BL/6 mice (18–22 g, Jackson Laboratories, Bar Harbor, ME, USA) were utilized. Animals were assigned unique study numbers via ear tags and cage cards and were randomly assigned to treatment or control groups (*n* = 5 per group) using a weigh-stratified randomization procedure. All animals

were quarantined and acclimatized for 7 days prior to study initiation and were housed in a 12:12 light/dark cycle with food and water provided *ad libitum*. Mice were infected intravenously with $10^4$ luciferase-expressing *P. berghei* sporozoites (ANKA strain) via tail vein injection on Day 0. Drugs were dissolved in PEG-400 and administered orally once per day on Day -1, Day 0, and Day 1. Liver-stage efficacy was determined by assessing the percent reduction of bioluminescence signals emitted in the liver area at 24 and 48 h post-infection, while the early blood-stage efficacy was assessed at 72 h by examining signals emitted from the whole body of the animal. Infection reduction was assessed by comparing the bioluminescence signals emitted by treated groups with signals emitted by the vehicle control-treated group at that time point. Later blood-stage parasitemia was determined via flow cytometry. In this model, an animal is considered cured if the liver-stage bioluminescence signal is below the IVIS limit of detection at 48 h post-infection and blood-stage parasitemia is below the flow cytometer limit of detection at 31 days post-infection. Long-term blood-stage efficacy was evaluated starting on Day 6 post-infection and continued for up to 31 days using flow cytometry.

## Animals and ethical statements at Walter Reed Army Institute of Research (WRAIR)

The animal protocols for research conducted for liver-stage efficacy and PK studies at WRAIR were reviewed and approved by the WRAIR/Naval Medical Research Command (NMRC) Institutional Animal Care and Use Committee (IACUC) and received formal oversight and approval from the U.S. Army Medical Research and Development Command (USAMRDC) (approved March 27, 2024). The experiments reported herein were conducted in compliance with the Animal Welfare Act and per the principles set forth in the "*Guide for the Care and Use of Laboratory Animals*," Institute of Laboratory Animals Resources, National Research Council, National Academy Press, 1996. Research was conducted in an AAALAC International-accredited facility in strict accordance with the *Guide for the Care and Use of Laboratory Animals* (NRC Publication, Eighth edition)[75]. Department of Defense (DoD) regulations, and the Animal Welfare Act. In compliance with DoD and institutional security policies, specific internal protocol numbers are mainained by the WRAIR Office of Animal Care and Compliance and are available for independent verification by the Editorial Office upon request.

## In vitro drug combination studies against blood-stage *P. falciparum* and liver-stage *P. cynomolgi*

For in vitro T111/TQ drug combination studies, blood-stage and liver-stage drug susceptibility tests were performed as described above, using a fixed-ratio platform for drug combinations[42,43,80]. For the asexual blood-stage assays, parasites were maintained in de-identified human A⁺ red blood cells purchased from Lampire Biological Laboratories (Pipersville, PA, USA). Briefly, drugs were diluted in fixed ratios (4:1, 3:2, 2:3, and 1:4) of starting concentrations predetermined to generate well-defined concentration response curves. Intrinsic dose-response curve for each drug alone and four different fixed-ratio combination dose-response curves yielded corresponding $IC_{50}$ values. The fractional inhibitory concentrations (FICs) were then calculated by the following formulas: FIC (A) = $IC_{50}$ of drug A in combination/$IC_{50}$ of drug A alone; FIC (B) = $IC_{50}$ of drug B in combination/$IC_{50}$ of drug B alone; ΣFIC index = FIC (A) + FIC (B). The isobolograms were constructed by plotting a pair of FICs for each combination. An interpretation of a straight diagonal line (ΣFIC index = 1) on the isobologram indicates an additive effect between two drugs. A concave curve below the line (ΣFIC index < 1.0) indicates synergy of the combination, while a convex curve above the line (ΣFIC > 1.0) indicates antagonism. All isobolograms were generated using Microsoft Excel (version 16.66.1).

### Ethical statement for human red blood cells used in blood-stage susceptibility test

The use of commercially available, de-identified human red blood cells for these in vitro assays was determined to be exempt from IRB oversight by the VA Portland Health Care System IRB, as the research did not involve "human subject" as defined by 45 CFR 46.102(f).

### In vivo drug combination studies against blood-stage *P. yoelii* and liver-stage *P. berghei* in mice

In vivo antimalarial efficacy studies were performed as described above. In the blood-stage 4-day suppression model, the effect of T111 on TQ oral efficacy was conducted by comparing dose-response curves of TQ alone and in the presence of sub-therapeutic oral doses of T111, and the potentiation effect was expressed as the ratio of $ED_{50}$ values for TQ and in the presence of T111. In the blood-stage single-dose cure model, various oral dose combinations of T111 with TQ were administered in mice 24 h post infection, and corresponding curative effects (no parasites detected) were determined on Day 28. For liver-stage in vivo oral efficacy studies, T111 alone, TQ alone, and sub-therapeutic combinations were administered to mice on Days −1, 0, and +1 after parasite inoculation. Parasite loads in the liver at 24 and 48 h were determined for liver-stage protection, and blood-stage cure was determined on Day 31.

### In vitro metabolic stability

The half-lives ($t_{1/2}$) of T111 in liver microsomes (HLM, MLM and RLM) were determined using our well-established methods[48,49]. Sample stocks at 20 μM in DMSO were diluted to a final concentration of 1.0 μM with a mixture containing 0.5 mg/mL prewarmed pooled human or mouse or monkey liver microsomes (BD Gentest), 1.3 mM NADP (Sigma), 3.3 mM $MgCl_2$ (Sigma), and 100 mM potassium phosphate buffer (pH 7.4) using a TECAN Genesis robotic liquid handler. The reaction was started with the addition of 1 U/mL glucose-6-phosphate dehydrogenase (G6PD). The mixture was incubated on a shaking platform at 37 °C, and aliquots were taken and quenched with the addition of an equal volume of cold acetonitrile at 0, 10-, 20-, 30-, and 60 min. Samples were centrifuged at 37,000 rpm for 10 min at 20 °C to remove debris. Sample quantification was carried out by LC-MS, and the metabolic half-life ($t_{1/2}$) was calculated by log plots of the total ion chromatography area remaining.

### In vivo pharmacokinetic (PK) studies

PK studies were conducted using well-established methods[21,31,50]. Four to five weeks old male and female outbred ICR-CD1 mice (Charles River Laboratories. Inc. Raleigh, NC) weighing 23–35 g were used for PK studies. Animals were assigned unique study numbers via ear tags and cage cards and were randomly assigned to treatment or control groups ($n = 3$ per group) using a weigh-stratified randomization procedure. All animals were quarantined and acclimatized for 7 days prior to study initiation and were housed in a 12:12 light/dark cycle with food and water provided *ad libitum*. T111 was formulated in sterile water containing 5% ethanol, 5% CRM, and 0.5% HECT, and given orally using intragastric administration. A single 40 mg/kg dose of T111 was administered to three mice at each experimental endpoint. At each time point, a terminal whole blood sample was collected by cardiac puncture using heparinized syringes (Heparin, Hospira, Lake Forest, IL), while mice were under anesthesia in accordance with the approved IACUC protocol. Following the separation of appropriate aliquots, plasma was obtained from the whole blood via centrifugation at 16,200 rpm for 10 min at 4 °C. Mice were humanely euthanized as described in the protocol, after which the liver samples were also collected and weighed at each experimental time point. All tissue samples were immediately preserved on dry ice and later stored at −80 °C until analytical work was performed. Drug concentrations were determined for each sample taken from animals dosed with the test

compound. PK parameters such as time to maximum plasma concentration ($T_{max}$), maximum plasma concentration ($C_{max}$), the apparent elimination half-life ($t_{1/2}$), area under the curve (AUC), etc., were obtained using noncompartmental analysis via the Phoenix WinNonlin software package (version 6.4; Pharsight Corp., Mountain View, CA).

### In vitro cytotoxicity assay

Cytotoxicity was evaluated in human HepG2 liver cells, using an MTT assay[52]. Briefly, HepG2 cells were cultured in complete Dulbecco's modified Eagle's medium (DMEM; Gibco-Invitrogen, No. 11995-065) prepared by supplementing DMEM with 10% heat inactivated FBS (Gibco-Invitrogen, No. 16000-036), 100 units/mL penicillin, and 100 μg/mL streptomycin. Cell viability was assessed using trypan blue. Ninety-six-well plates were seeded with $2 \times 10^4$ cells per well and incubated at 37 °C overnight in a humidified 5% $CO_2$ atmosphere. Test compound plates were prepared with an INTEGRA assist plus station and 96-well plates containing 11 duplicate 2-fold serial dilutions of test compound suspended in DMSO. The diluted test compound was then added to the corresponding well preseeded with HepG2 cells. Cells were exposed to test compound concentrations ranging from 200 μM to 0.2 μM for 24 h. Next, the media and test compounds in each well were removed and plates were then added with fresh drug-free medium and continued to incubate for an additional 24 h. After incubation, 20 μL of a 0.1 mg/mL solution of resazurin diluted in PBS was added to each well, and all plates were subsequently incubated for 3–4 h in an incubator. Absorbance was determined with the Molecular Devices SpectraMax plate reader. Three independent experiments, each in duplicate, were performed, and the $IC_{50}$ values were determined using GraphPad Prism software (Version 10) using the non-linear regression equation (sigmoidal dose-response, variable slope).

### hERG channel inhibition assay

The hERG assay was performed at Eurofins Panlabs, Inc., St. Charles, MO, United States. T111 was tested using the standard hERG potassium channel assay[53] at six concentrations from a top concentration of 100 μM. Verapamil was used as a positive control.

### Ames assay

Mutagenicity was assessed using the Ames assay[54] (EBPI, Ontario, Canada) against *S. typhimurium* TA100 and TA98, with and without S9 activation. The tester strain *S. typhimurium* (TA100 and TA98) cultures were inoculated from the lyophilized pellets and grown in nutrient broth provided by EBPI. Each inoculated vial (5 mL volume with antibiotics) was placed overnight in the incubator at 37 °C. The positive controls for the assay were 1.0 μg/μL 2-aminoanthracene (2-AA) for incubations with the S9 mix and 5 μg/mL sodium azide ($NaN_3$, used for TA100) and 300 μg/mL 2-nitrofluorene (2NF, used for TA98) for incubations without the S9 mix. The tester strains (5 μL) were mixed with 2.5 mL reaction mixture and, where appropriate, 2.0 mL of S9 mixture, T111 (200 μL/reaction), was then added to each reaction tube to give a final volume of 20 mL. An aliquot (0.2 mL) of the mixture was then added to each well of a 96-well plate. The 96-well plates were covered, sealed in zip-lock bags, and incubated for 5 days at 37 °C. Plates were scored visually, and the positive wells against background mutations were statistically analyzed.

### Eryptosis assay

In vitro evaluation of eryptosis as a marker for induced hemolytic toxicity was performed using de-identified human blood samples obtained from the Clinical Trial Center at WRAIR, using established methods[55,81,82]. Fresh Li-heparin-anticoagulated blood samples was centrifuged and the platelet and leukocyte-containing supernatant was removed. Erythrocytes were incubated with TQ or T111 in Ringer solution at 37 °C for 48 h. Flow cytometry was employed to estimate stimulators of eryptosis: phosphatidylserine exposure at the cell surface from annexin-V-binding,

reactive oxygen species (ROS) formation from 2′,7′-dichlorodihydro-fluorescein diacetate (DCFDA) dependent fluorescence, and cell volume from forward scatter. Annexin-V is a small protein that has a high affinity for phosphatidylserine and annexin-V binding is used to measure the presence of phosphatidylserine on the cell membrane surface and subsequently the eryptotic activity of the cell.

## Annexin-V-binding assay for detection of phosphatidylserine translocation

After incubation under the respective experimental condition, cell suspension was washed in Ringer solution containing $CaCl_2$ and then stained with Annexin-V-FITC at 37 °C for 15 min under protection from light. The annexin-V-abundance at the erythrocyte surface was subsequently determined on a FACS Calibur. Annexin-V-binding was measured at an excitation wavelength of 488 nm and an emission wavelength of 530 nm. A marker (M1) was placed to set an arbitrary threshold between annexin-V-binding cells and control cells. The same threshold was used for untreated and drug treated erythrocytes.

## Reactive oxygen species (ROS) detection assay

Oxidative stress was determined utilizing DCFDA. After incubation, a 150 μL suspension of erythrocytes was washed in Ringer solution and stained with DCFDA in Ringer solution containing DCFDA at a final concentration of 10 μM. Erythrocytes were incubated at 37 °C for 30 min in the dark and washed two times in Ringer solution. The DCFDA-loaded erythrocytes were resuspended in Ringer solution and ROS-dependent fluorescence intensity was measured at an excitation wavelength of 488 nm and an emission wavelength of 530 nm on a FACS Calibur. Subsequently, the geomean of the DCFDA dependent fluorescence was determined.

## Quantification of cell size

Erythrocyte volume was estimated utilizing forward scatter (FSC) as a measure of light diffraction. After incubation in the presence or absence of the test compounds, the erythrocyte suspension was washed and resuspended in Ringer solution. Cells were identified and gated based on their light-scatter characteristics (FSC-linear vs. SSC-linear) on a FACS Calibur to exclude debris and cellular aggregates. For the analysis of cell volume, an FSC-linear histogram was generated from the gated population. The FSC gain was adjusted to fix the geometric mean (Geo Mean) of the untreated control erythrocytes at approximately 500 arbitrary units (AU). The same gain and threshold settings were maintained for both untreated and drug-treated normal and G6PD-deficeinet erythrocytes. Shifts in the Geo Mean of the FSC signal were subsequently determined to quantify changes in cell volume relative to the baseline control.

## Ethical statement for use of human erythrocytes in the eryptosis assay

The study protocol was approved by the WRAIR Institutional Review Board in compliance with all applicable Federal regulations governing the protection of human subjects. Specifically, the study protocol for the use of volunteer-derived human erythrocytes (normal and G6PD-deficient RBCs) was reviewed and approved by the Ethics Committee of the WRAIR under specimen collection protocol #2567.04, titled ′ Blood Collection for G6PD Antimalarial Assay′ (version 1.0, approved 21 December 2018). Written informed consent was obtained from all participants prior to collection in accordance with institutional and federal regulations (45 CFR 46). All blood products were provided as de-identified specimens and assigned unique study identifiers prior to laboratory analysis.

## In vivo toxicology and toxicokinetic (TK) studies in rats

In vivo toxicology and TK studies in rats were performed at SRI International, using the well-established methods[83–85]. Rats were

housed in microisolator cages with hardwood chip bedding (Sani-Chips; Envigo Teklad) and maintained on a 12 h light/dark cycle. Environmental conditions were controlled at 20–26 °C (68–79 °F) with 30–70% relative humidity. Animals were group-housed, except where individual housing access to a certified global 18% protein rodent diet (Envigo Teklad) and purified water. T111 was dissolved in PEG-400 and continuously stirred in a heated water bath between 85 and 98 °C for 1–4 h, until a clear solution or a homogeneous formulation was obtained. Male and female Sprague Dawley outbred rats were randomly assigned to treatment and control groups using a simple randomization procedure to ensure a balanced distribution of baseline weights across all cohorts. For the MTD study, each group of two male and two female Sprague Dawley rats were administered a single dose of 30, 100, 200 or 400 mg/kg T111 by oral gavage and the animals were monitored immediately post dose, 2–4 h and 24 h post dose for clinical signs, with body weights recorded pre-dose and on Day 1 for each dose level group. For the 7-day repeat dose study, each group of eight male and eight female Sprague Dawley rats were administered a single oral dose of T111 (25, 100 or 400 mg/kg) or vehicle control daily for 7 consecutive days and evaluated for body weight changes (pre-dose, on Day 8 and 14, and prior to euthanasia), clinical observations (immediately, 2–4 h post dose, and daily on Day 1 to 7), plasma drug levels (on Day 1 and 7), clinical pathology (hematology, clinical chemistry on Day 8 and 14), organ weights (brain, heart, kidney, liver, spleen; on Day 8 and 14), and gross and microscopic examination of the selected tissues (brain, cecum, colon, duodenum, heart, ileum, jejunum, kidneys, liver, lungs, lymph nodes, rectum, spleen, forestomach, glandular stomach; on Day 8 and 14). For TK studies, blood was collected from animals at 6 time points post dose on Day 1 (0.5, 1, 2, 4, 8, and 24 h) and on Day 7 (0.5, 2, 8, 24, 48, and 96 h). Drug levels were determined in collected plasma samples using bioanalytical method developed at SRI. The plasma drug level data were analyzed using Phoenix Win-Nonlin (version 8.3) software to perform noncompartmental data analysis.

## Animals and ethical statements at SRI International (SRI)

All rat safety and TK studies were conducted at SRI International (Menlo Park, CA, USA) in an AAALAC-accredited facility. All procedures were reviewed and approved by the SRI International Institutional Animal Care and Use Committee (IACUC Protocol #02006) and received formal oversight and approval from the U.S. Army Medical Research and Development Command (USAMRDC) Animal Care and Use Review Office (ACURO Protocol #PR210491.e002) on September 21, 2023. Studies were performed in strict accordance with the *Guide for the Care and Use of Laboratory Animals* (NRC Publication, Eighth edition)[75] and the U.S. Department of Agriculture (USDA) Animal Welfare Act.

## Drug resistance selection studies

Drug resistance selection studies were performed by culturing *P. falciparum* Dd2 parasite under incremental and continuous drug pressure of T111, using a well-established method[30,86]. Four independent cultures of *P. falciparum* Dd2 were maintained in the presence of T111 (0.1, 0.5, or 2 nM), with a non-drug exposed control grown in parallel. The concentration of T111 was incrementally increased when parasites replicated at the same rate as the untreated parent line for three life cycles. Selection experiments were completed when T111 pressure reached 1 μM, at which point the culture was cryopreserved. Cryopreserved parasites from specific T111 concentrations were thawed and cloned by limiting dilution. DNA extractions were performed on clonal parasites using the QIAamp DNA extraction kit (QIAGEN Inc, Valencia, CA) or ZYMO Quick gDNA Miniprep kit (Zymo Research Corp, Irvine, CA). For sequencing of the open reading frame of *pfcytb*, PCR was performed with genomic DNA using forward and reverse flanking primers for *cyt b* primer 1: 5′-TTCCTGATTATCCAGACGCT-3′

and primer 2: 5′-TGTTCCGCTCAATACTCAGA-3′. PCR products were subjected to enzymatic cleanup with ExoSAP-IT (Thermo Fisher Scientific) and sent for Sanger sequencing (Eurofins Genomics, Louisville, KY) with PCR primer 2 paired with an internal forward primer: 5′-GAGTTATTGGGGTGCAACTG-3′ and an internal reverse primer: 5′-CACTCACAGTATATCCTCCACA-3′. Sequences were assembled using 4Peaks software (Nucleobytes, Amsterdam, Netherlands) and aligned with Clustal Omega (EMBL-EBI). For sequencing of the open reading frame of *pfdhodh*, PCR was performed with genomic DNA using forward primer: 5′-GTGATAGATAGCTCCAGTCGATTTC-3′ and reverse primer: 5′-TTTGCGCACTTATGTGTCGCCC-3′. PCR products were prepared for Sanger sequencing and analyzed as described above, except using two forward primers: 5′-GCTATTAATGTAAGCTCCC-3′ and 5′-CCATTCGGTGTTGCTGC-3′ and with two reverse primers: 5′-GCATTACCCGTTTGGCCCCTTGGGG-3′ and 5′-GGAGCTTACATTAATAG-3′. Newly generated T111-resistant *P. falciparum* clones and parental strains described in this study are available from the corresponding author upon reasonable request for non-commercial research purposes, subject to standard Material Transfer Agreements (MTA).

### Genomic DNA isolation, sequencing, and whole genome sequencing (WGS) analysis

T111 mutants and the parental strain were evaluated with Illumina whole genome sequencing. Genomic DNA was isolated from six clones resulting from drug selection experiments (Qiagen DNeasy) and concentrated per the manufacturer's instructions (Zymo Research Corporation Genomic DNA Clean & Concentrator™ Kit-25). Libraries for Illumina short-read sequencing were prepared and sequenced (Illumina NovaSeq X Plus, Novogene). Genomic sequence data were aligned to the Dd2 genome (version 51 from PlasmoDB[87,88]), sorted with samtools (v1.11)[89], duplicates marked with picrad (v2.25.0)[90], variants called with freebayes (v1.3.5), filtered for quality, and annotated with SnpEff (v5.0). Single-nucleotide variants (SNVs) in protein-coding regions identified in the parental Dd2 strain were subtracted from those identified in each T111-resistant strain to identify those unique to each resistant strain. To identify SNVs implicated in T111 resistance, SNVs present in less than 90% of the reads of a particular strain were excluded.

### *P. falciparum cyt b* modeling and autodocking calculations

Automated docking simulations were conducted using the AutoDock 4.0 program suite[91]. The sequence of *P. falciparum cyt b* is highly homologous to yeast *cyt b* and thus the crystal structure of yeast *cyt b* (PDB ID 4PD4) was used for ligand docking studies with corresponding residues replaced with those of *P. falciparum cyt b* around the ATV binding pocket. The 3D structures of compounds T63, T111, ELQ300 and ELQ400 were modeled based on molecular formulas in PubChem database. The crystal structures of ATV, MYX, or other model structures were roughly placed in the binding pocket as described in the yeast *cyt b*-ATV complex, which was used as the initial starting point to minimize the complexes using the Lamarckian genetic algorithm. The crystal structure of bovine *cyt b*-MYX complex (PDB ID 1SQP) was also used in docking study as a reference. In all cases, a grid box size of $60 \times 60 \times 60$ points with a grid-spacing of 0.375 Å was implemented, which was centered on the ligand to cover the whole pocket. Thirty docked structures, i.e., 30 runs, were generated by using genetic algorithm searches. A default protocol was applied, with an initial population of 50 randomly placed individuals, a maximum number of $2.5 \times 10^5$ energy evaluations, and a maximum number of $2.7 \times 10^4$ generations. A mutation rate of 0.02 and a crossover of 0.8 were used. Results differing by less than 1.0 Å in positional root-mean-square deviation (RMSD) were clustered together. A ligand conformation with the lowest docking energy and catalytically binding to the protein was considered to be docked.

### Drug susceptibility testing for cross-resistance profile

Drug susceptibility assays against *P. falciparum* strains were performed using a SYBR Green based assay[44]. D6 and Dd2 were obtained from MR4. Tm90-C2B was obtained from WRAIR. Dd2-D1$^{122L}$ and ELQ-300 were obtained from VAPORHCS. D10yDHODH was obtained as a courtesy from Dr. Akhil Vaidya at Drexel University. *P. yoelii* ATV$^r$ and ELQ-300$^r$ clones were obtained from VAPORHCS.

### Statistical analysis and sample size determination

Sample sizes for in vivo efficacy and PK studies were determined by a priori power analysis using G*power software (version 3.1.9.7). Based on the difference between two independent means observed in previous experiments (Cohen's $d = 2.34$), a one-tailed t-test ($\alpha = 0.05$, power = 0.80) was used to justify a group size of $n = 4$ for blood-stage efficacy and $n = 5$ for liver-stage efficacy. All statistical comparisons were performed in GraphPad Prism (versions 8.0–10) or Microsoft Excel (version 16.66.1). For eryptosis assay (Fig. 4), significance was determined using two-way ANOVA. For gametocyte inhibition assays, the non-parametric Kruskal-Wallis with Dunn's post0hoc test was used. Mosquito infection intensity (SMFA) was analyzed using the Friedman test. Tarsal contact assay was analyzed with two-tailed Mann-Whitney test, while infection prevalence was compared using two-tailed Fisher's exact tests. Data are presented as mean ± SEM or mean ± SD as indicated in the figure legends. $p$ values < 0.05 were considered statistically significant.

### Reporting summary

Further information on research design is available in the Nature Portfolio Reporting Summary linked to this article.

## Data availability

Source data are provided with this paper. The raw and processed data generated in this study have been deposited in the Figshare database under accession code 31825489. The Whole Genome Sequence (WGS) data generated in this study have been deposited in the NCBI BioProject database under accession code PRJNA1417775, which contains the 7 individual SRA records (SRX32021460–SRX32021466) corresponding to the *P. falciparum* Dd2 parental and derived mutant lines analyzed in this study. No data in this study are under restricted access or protected by data privacy laws.

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

## Acknowledgements

This work was supported by NIH/NIAID (award numbers AI158533 to J.X.K.; AI075045 and AI139179 to P.J.R.; AI181593 to L.C.; and AI127338 to M.T.F.), DOD/PRMRP (award number W81XWH2210494 to J.X.K.), and US Department of Veterans Affairs (VA Merit Review Award BX004522 to J.S.D., and BX005674 to J.X.K.). The National Science Foundation (NSF) is acknowledged for support of the BioAnalytical Mass Spectrometry Facility at Portland State University (MRI1828573). This research was also supported in part by an appointment to the Department of Defense (DoD) Research Participation Program administered by Oak Ridge Institute for Science and Education (ORISE) through an interagency agreement between U.S. Department of Energy (DOE) and the DOD. ORISE is managed by ORAU under DOE contract number DE-SC0014664. In addition, this work was supported by the NIH Distinguished Scholars

Program and the Intramural Research Program of the Division of Intramural Research (AI001250-01), NIAID/NIH to J.V.R. We acknowledge Dr. Amrendra Kumar for assistance with HRMS and HPLC analyses, and Dr. David Peyton for valuable scientific discussions. Procurement of ART-resistant clinical isolates was supported by German Research Foundation (DFG; ref no. GRK2046, and GRK2290). Material has been reviewed by the WRAIR. There is no objection to its presentation and/or publication. The opinions or assertions contained herein are the private views of the authors, and are not to be construed as official, or as reflecting true views of the Department of the Army or the Department of Defense.

## Author contributions

P.K., and R.A.D. contributed to drug design and synthesis; Y.Li, and X.Z. contributed to asexual blood-stage in vitro and in vivo testing; S.K. contributed to synergy analysis; J.C. and R.A.C. contributed to drug resistance selection studies and ex vivo testing; A.J.B., K.N., L.G., and J.Mu contributed to drug resistance selection studies; J.L. contributed to docking modeling analysis; D.C., M.S.M., X.J., W.E.D., K.K., S.M., P.J.L., C.B., J.D., C.V., K.P., H.T.D., K.M., S.L., M.L.M., B.S.P., G.C.C., and A.R. contributed to liver-stage in vitro and in vivo testing, eryptosis assay, and DMPK profiling; A.C.O., and L.C. contributed to in vitro drug susceptibility test against ART-resistant parasites; T.Q., D.A.S., M.T.F., F.P.M., W.V.L., and J.M.N contributed to ART-resistant clinical isolates testing; H.K., and J.V.R. contributed to gametocyte testing; M.O., O.K., and P.J.R. contributed to ex vivo testing; C.D., C.G., Y.Liu, A.A., and J.Mirsalis contributed to in vivo toxicology in rats; S.N.F., G.I.M., and C.D.G. contributed to SMFA assay; A.S.P., A.N., and F.C. contributed to tarsal contact assay; P.H.A., and J.S.D. contributed to genome sequencing and analysis; M.K.R. contributed to ELQ compound supplies and mutant lines; J.X.K. contributed to study design, blood-stage testing in vitro and in vivo. J.X.K., P.K., and P.J.R. contributed to manuscript writing, review and editing. This work is dedicated to the memory of our late colleague and co-author, Dr. Roland A. Cooper, whose profound passion and fundamental scientific contributions to malaria research were instrumental in the conception and completion of this study. His dedication to the field remains an inspiration to us all.

## Competing interests

The authors declare the following competing interests: P.K., R.A.D., and J.X.K. are listed as inventors on a United States provisional patent application filed by Portland State University to protect the intellectual property on the acridone prodrugs described in this study. All other authors declare no competing financial or non-financial interests.

## Additional information

**Papireddy Kancharla** [1] ✉, **Rozalia A. Dodean** [1,2], **Yuexin Li** [2], **Xiaowei Zhang** [2], **Sean Kelly** [2], **Jordan Charlton** [3], **Angely J. Binauhan** [3], **Kimberly Navarrete** [3], **Laurize Garcia** [3], **Jianbing Mu** [4], **Jinghua Lu** [5], **Diana Caridha** [6], **Michael S. Madejczyk** [6], **Xiannu Jin** [6], **William E. Dennis** [6], **Karl Kudyba** [6,7], **Sharon McEnearney** [6], **Patricia J. Lee** [6], **Cameron Blount** [6], **Jesse DeLuca** [6], **Chau Vuong** [6], **Kristina Pannone** [6], **Hieu T. Dinh** [6], **Kennedy Mdaki** [6], **Susan Leed** [6,7], **Monica L. Martin** [6], **Brandon S. Pybus** [6], **Geoffrey C. Chin** [6], **Anongruk Chim-Ong** [8], **Liwang Cui** [8], **Tarrick Qahash** [9], **Douglas A. Shoue** [9], **Michael T. Ferdig** [9], **Frank P. Mockenhaupt** [10], **Welmoed van Loon** [10], **Jules M. Ndoli** [11], **Heather Kudyba** [4], **Joel Vega-Rodriguez** [4], **Martin Okitwi** [12], **Oriana Kreutzfeld** [13], **Cristina Docan** [14], **Carol Green** [14], **Ying Liu** [14], **Aaron Agulay** [14], **Jon Mirsalis** [14], **Sarah N. Farrell** [15], **Alexandra S. Probst** [16], **Aaron Nilsen** [2,17], **P. Holland Alday** [2], **J. Stone Doggett** [2,18], **Geoffrey I. McFadden** [15], **Christopher D. Goodman** [15], **Flaminia Catteruccia** [16,19], **Philip J. Rosenthal** [13], **Michael K. Riscoe** [2,20], **Roland A. Cooper** [3], **Alison Roth** [6,7] & **Jane X. Kelly** [1,2] ✉

[1]Department of Chemistry, Portland State University, Portland, OR, USA. [2]Department of Veterans Affairs Medical Center, Portland, OR, USA. [3]Department of Natural Sciences and Mathematics, Dominican University of California, San Rafael, CA, USA. [4]Laboratory of Malaria and Vector Research, National Institute of Allergy and Infectious Diseases, National Institute of Health, Rockville, MD, USA. [5]Lab of Immunogenetics, Structural Immunology Section, National Institute of Allergy and Infectious Diseases, National Institutes of Health, Rockville, MD, USA. [6]Experimental Therapeutics Branch, Walter Reed Army Institute of Research, Silver Spring, MD, USA. [7]Integrated Pathogen Therapeutics Department, Walter Reed Army Institute of Research, Silver Spring, MD, USA. [8]Department of Internal Medicine, Morsani College of Medicine, University of South Florida, Tampa, FL, USA. [9]Eck Institute for Global Health, Department of

Biological Sciences, University of Notre Dame, Notre Dam, IN, USA. [10]Institute of International Health, Charité Center for Global Health, Charité-Universitaetsmedizin Berlin, Freie Universität Berlin and Humboldt-Universität zu Berlin, Berlin, Germany. [11]College of Medicine and Health Sciences, University Teaching Hospital of Butare, University of Rwanda, Huye, Rwanda. [12]Infectious Diseases Research Collaboration, Kampala, Uganda. [13]Department of Medicine, University of California San Francisco, San Francisco, CA, USA. [14]Biosciences Division, SRI International, Menlo Park, CA, USA. [15]School of Biosciences, The University of Melbourne, Parkville, VIC, Australia. [16]Department of Immunology and Infectious Diseases, Harvard TH Chan School of Public Health, Boston, MA, USA. [17]Chemical Physiology and Biochemistry, Oregon Health & Science University, Portland, OR, USA. [18]Department of Medicine, Oregon Health & Science University, Portland, OR, USA. [19]Howard Hughes Medical Institute, Boston, MA, USA. [20]Department of Microbiology and Molecular Immunology, Oregon Health & Science University, Portland, OR, USA. ✉e-mail: papiredd@pdx.edu; kellyja@ohsu.edu

