## [Peer Review file · Nature Communications]

Potent Acridone Antimalarial against All Three Life Stages of Plasmodium

Corresponding Author: Dr Jane Kelly

Version 0:

Reviewer comments:

Reviewer #1

(Remarks to the Author)

The manuscript describes the evaluation of an antimalarial drug candidate. The main selling points are: i) a picomolar activity against disease-causing asexual blood stages; ii) activity against other major life-cycle stages, including gametocytes, liver stages and insect stage, albeit with highly variable efficacies; iii) transmission blocking activity; iv) efficacy in a mouse malaria model system; and v) favorable safety profiles in acute and repeat toxicology studies in rats. The study features multiple assay systems to support the broad-spectrum efficacy and safety of the compound. The results of several different assay systems are shown to support the broad potency of the compound and its safety in relevant systems. However, some important data are either missing or contradictory. Therefore, further investigations are necessary to fully assess the therapeutic potential of the compound.

1. History of malaria. The opening sentence of the introduction should be corrected. Malaria is an ancient disease of humanity, with evidence suggesting it may have coexisted with early humans for over 2.4 million years, as indicated by findings such as Plasmodium-infected mosquitoes preserved in amber. Please revise the sentence to accurately reflect this.
2. Determination of DHA responsiveness. The standard growth inhibition assay is insufficient for evaluating artemisinin responsiveness, as it fails to capture the delayed clearance phenotype. The ring-stage survival assay (RSA) is a more appropriate and widely accepted method. It is recommended that RSA be used to benchmark the compound's activity against dihydroartemisinin (DHA).
3. Responsiveness of field isolates. The more than 150-fold variation in IC₅₀ values among field isolates is surprising and requires further explanation. This range could indicate early signs of resistance. It is recommended to perform genetic analysis on low-responder isolates, focusing at minimum on the *cyt b* and *pfhdhdh* loci. Whole genome sequencing (WGS) of representative isolates would be even more informative.
4. Discrepancy between in vitro and in vivo activity. The marked discrepancy between the in vitro activity (IC₅₀ values in the picomolar range) and the required in vivo curative dose in a mouse malaria model system (50 mg/kg) raises several questions that require further investigation. Possible explanation include: poor oral bioavailability of the compound, high metabolic turn-over in mice, poor responsiveness of *P. yoelii* due to altered uptake or metabolic stability of the compound. To address these critical issues the following experiments should be performed: i) PK in mice including determination of whether parasite killing is C_{max} or AUC driven; ii) metabolism of the compound using liver microsomes from mice and humans; iii) uptake and efflux kinetics of the compound by *P. falciparum* infected human erythrocytes and *P. yoelii* infected mice erythrocytes (Is uptake mediated by NPP?); iii) target validation in *P. yoelii* with regard to *cyt b* and *pfhdhdh*; and iv) stability of the compound in short term *P. falciparum* and *P. yoelii* cultures
5. Molecular targets of the compound. It is stated that *cyt b* and possibly *pfhdhdh* are targeted by the compound. It is unclear whether this statement is based on whole genome sequencing of resistant lines or on targeted sequencing of just the two loci. If the latter is the case, then other important resistance conferring mutations may have been missed. It is therefore recommended to determine the whole genome of some resistant line, possibly those that display different levels of responsiveness to evaluate evolution of resistance. It is further recommended to evaluate the contribution of the two resistance associated genes independently using e.g., CRISPR Cas9 genome editing technology.
6. Inhibition of *cyt b* by the compound. Molecular dynamics simulations should be performed to better understand the interaction of the compound with *cyt b* at the molecular level and how mutations in the protein cause resistance. Additionally, in vitro binding studies using purified *cyt b* protein would help clarify the mechanism of inhibition and the impact of resistance-associated mutations on binding kinetics.
7. PK/PD model in humans. The development of a human PK/PD model is critical to predict clinical efficacy and inform dose

selection. The authors should construct such a model to define the relationship between dosage, efficacy, and safety in humans.

Reviewer #2

(Remarks to the Author)

1. A consistent number of significant figures should be used for the IC50 and EC50 data.
2. The authors should generate in vitro ADME/PK data or discuss existing in vitro ADME/PK data for their lead compound T111 and its prodrug.
3. Why did the authors not assess if T111 also had activity against hypnozoites?

Reviewer #3

(Remarks to the Author)

Review of "Potent Acridone Antimalarial against All Three Life Stages of Plasmodium" by Kancharla et al.

Brief summary of the most noteworthy results:

The authors of this manuscript have performed extensive experimental testing of the potential new antimalarial compound T111 using different malaria species and different stages of the life cycle (blood stage, gametocytes, mosquito stages, liver stages), using a combination of in vitro and in vivo experiments. All the malaria species in all the stages of the life cycle were eliminated by this compound, even hypnozoites in radical cure mode (the most difficult stage to treat) were susceptible. Several drug resistant *P. falciparum* strains were tested and almost all of those were still susceptible to treatment with T111. Induction of resistance was difficult and could be pinpointed to a set of mutations in CytB Q0 site. Toxicity testing showed hardly any toxicity issues for this compound, making the compound very promising. The experimental design is good and execution of the experiments is sound. New antimalarials are desperately needed and this compound could be beneficial for a lot of people. Hopefully this product will proceed to clinical testing and can make a difference in the combat against malaria in all its shapes and forms.

Validity

Relevant numbers of replicates have been performed for all of the experiments, the data look consistent and standard deviations are within the expected range.

Significance

The combat against malaria could benefit from the use of this compound in a clinical setting. This set of experiments is showing the potential of T111 as new antimalarial in several in vitro/vivo models. Especially interesting is the observation of activity of T111 against hypnozoites in a radical cure setting, as this would be a new scaffold replacing the much debated 8 aminoquinolines. Or it could be given as a combination drug thereby lowering the tafenoquine dose drastically.

Data and methodology

The data are presented clearly and the number of biological and technical replicates are sufficient.

It surprised me that, although many *P. falciparum* strains carrying various drug resistance mutations were tested in blood stage assays, no experiments were performed testing the efficacy on *P. vivax* blood stages (or *P. cynomolgi* as a proxy for *vivax*).

I also have some remarks on the methodology of the in vivo testing of the synergy between tafenoquine/T111. The model system used (*P. yoelii* in mice for blood stage testing and *P. berghei* in mice for liver stage testing). To my knowledge tafenoquine alone is not used to treat blood stages because of the high dose needed to eliminate all gametocytes. Rather than looking at clearance of the blood stage alone, it would have been desirable to test the effect of the drug combinations on transmission. Moreover, the liver stage testing of tafenoquine and T111 in *P. berghei* infected mice does not include testing of T111 against hypnozoites, as this stage of the life cycle is not observed in this model. Although the in vitro activity against hypnozoites is promising, it is not supported by any in vivo evidence.

Analytical approach

In general, the analyses are clear and straightforward.

I only wonder about the Kruskal-Wallis test used for significance calculation in fig 1E, to compare oocysts of untreated vs oocysts of treated. Does not seem to me that a multiple-comparison test is needed and significance also for the lower dose would probably be shown if the statistical analysis would include a one-way ANOVA test (BUT I'm not a statistician!).

Suggested improvements

This manuscript is well written, contains a huge amount of data, which are presented very clearly and without any doubt T111 is a promising compound for progression into clinical testing.

The inclusion of in vivo testing of T111 as a radical cure would test the hypothesis of the activity of this compound against established hypnozoites. For this I would propose to include a set of experiments in which both prophylactic and radical cure testing of T111 (either by itself or in combination with tafenoquine) is performed on either humanized mice/*P. vivax* or rhesus monkeys/*P. cynomolgi*.

I have categorized other minor improvements or questions according to the sequence in the manuscript

Regarding blood stage testing

In Fig 1B the authors show that there is quite a lot of variation in the IC50 measured in the clinical isolates, although the

IC50's are still low. Would that be an indication of the presence of parasites already carrying one or two mutations in the CytBQ0 site?

Regarding the in vivo testing against *P. yoelii* blood stages (page 6 and fig 1C), I wondered how it was decided upon the treatment doses (40 and 50 mg/kg) tested in the mouse experiments. Linked to that, were there any Pk/PD tests performed beforehand using these doses? I'm asking because it looks like the difference in treatment results between the 40 and 50 mg/kg are quite striking, so I wondered if this can be explained somehow.

Also, the authors state that there is no male-female difference in Pk/PD parameters, but it looks to me that the T1/2 is shorter in females (table 3), can you check if there is a statistically relevant difference in T1/2?

In the text it is written 'long half-life' (p17). Do the authors think 15-18hr half-life is long? If you compare this to the T1/2 of other antimalarials (T1/2 of chloroquine is 20-40 days, T1/2 of tafenoquine is 14 days T1/2 of atovaquone is 5-6 days), the T1/2 is not very long.

Gametocyte stages

Suggestion for the layout of the paper: It would be more logical to put the mosquito stage testing after the gametocytes if the manuscript follows the mosquito life cycle, which it did up to this point.

Liver stages

The activity of T111 is compared to the reference drugs Atovaquone and tafenoquine. So why is maduromycin in the table? The authors also state in the discussion that atovaquone is not active against hypnozoites, but the IC50 of atovaquone against hypnozoites is quite low. That is contradictory.

Does T111 eliminate the hypnozoites from the cultures completely or do the authors still see tiny parasites after prophylactic or radical cure treatment?

Mosquito stages

In the tarsal contact tests, reduced oocyst numbers were seen. But do these still result in sporozoites? And if so, are the sporozoites infective?

Was the transmission blocking effect tested only by adding the compound to infected blood? I think these kinds of tests are suitable as a first screening, but feeding on infected/treated animals would be additionally informative with regards to transmission testing.

Fig 1E what is the oocyst load of the untreated control?

Synergy experiments

The synergy between T111 and tafenoquine is obvious. Inhibition of the Q0 by T111 increases the parasite's sensitivity for tafenoquine. Does this also help to identify the mode of action of tafenoquine?

Fig 2C Are the T111 alone curves not included?

I found a typo in page 29

Header: Drug subspeciality testing for cross-resistance profile.

I assume the authors mean susceptibility here

With kind regards

Anne-Marie Zeeman

Version 1:

Reviewer comments:

Reviewer #1

(Remarks to the Author)

The authors have substantially strengthened the revised manuscript, significantly improving both the depth and breadth of the data supporting their central claim. In particular, the additional experimental evidence convincingly supports the conclusion that T111 represents a previously unexploited chemotype with stage-transcending antimalarial activity, a distinct resistance mechanism, and a favorable safety profile.

My initial recommendations have been comprehensively addressed, and the revisions have enhanced the impact of the study.

Reviewer #2

(Remarks to the Author)

NONE

Reviewer #3

(Remarks to the Author)

I have read the rebuttal and the revised manuscript. All the points have been addressed. I have no further comments

Potent Acridone Antimalarial against All Three Life Stages of Plasmodium NCOMMS-25-44081-T

Point-by-Point Response to Reviewers Comments

Reviewer #1 (Remarks to the Author):

The manuscript describes the evaluation of an antimalarial drug candidate. The main selling points are: i) a picomolar activity against disease-causing asexual blood stages; ii) activity against other major life-cycle stages, including gametocytes, liver stages and insect stage, albeit with highly variable efficacies; iii) transmission blocking activity; iv) efficacy in a mouse malaria model system; and v) favorable safety profiles in acute and repeat toxicology studies in rats. The study features multiple assay systems to support the broad-spectrum efficacy and safety of the compound. The results of several different assay systems are shown to support the broad potency of the compound and its safety in relevant systems. However, some important data are either missing or contradictory. Therefore, further investigations are necessary to fully assess the therapeutic potential of the compound.

1. History of malaria. The opening sentence of the introduction should be corrected. Malaria is an ancient disease of humanity, with evidence suggesting it may have coexisted with early humans for over 2.4 million years, as indicated by findings such as Plasmodium-infected mosquitoes preserved in amber. Please revise the sentence to accurately reflect this.

Response: We are appreciative of this note and the opening sentence has been revised accordingly.

2. Determination of DHA responsiveness. The standard growth inhibition assay is insufficient for evaluating artemisinin responsiveness, as it fails to capture the delayed clearance phenotype. The ring-stage survival assay (RSA) is a more appropriate and widely accepted method. It is recommended that RSA be used to benchmark the compound's activity against dihydroartemisinin (DHA).

Response: The RSA has proved problematic for ex vivo assays of fresh African parasites, and in fact standard dose response curves, as included for our ex vivo assays of parasites from Uganda and Burkina Faso, are better indicators of ART-resistance (see PMID 40783405). However, based on the reviewer's appropriate suggestion, T111 was tested in vitro against two *P. falciparum* clinical isolates, ART-sensitive SP045 and ART-R SP060, using the recently described Growth, Resistance, and Recovery Assay (GRR), an optimized higher-throughput version of the RSA that allows us to generate a dose response curve (11 replicated RSAs per drug/parasite combinations). As expected, DHA showed reduced susceptibility in the ART-R mutant, but T111 was far more potent than DHA and maintained comparable activity against both ART-sensitive and ART-resistant parasites. (Figure 1A and 1B of the revised manuscript).

3. Responsiveness of field isolates. The more than 150-fold variation in IC₅₀ values among field isolates is surprising and requires further explanation. This range could indicate early signs of

resistance. It is recommended to perform genetic analysis on low-responder isolates, focusing at minimum on the *cyt b* and *pfhdhdh* loci. Whole genome sequencing (WGS) of representative isolates would be even more informative.

Response: We appreciate the reviewers reasonable concerns, but in our experience, *ex vivo* assays routinely include uncommon outliers with unusually high or low IC₅₀s, likely due to compound solubility limitations, relatively poor parasite growth, human error, or other factors that cannot be corrected, as assays necessarily cannot be repeated. We present all data, but appreciate that, as has been the case with a few dozen other compounds studied by our group (see PMID 34339273, 36094216, 37158739), uncommon outliers may be misleading. With this understanding, because resources are not available for the requested genomic studies, and most importantly because all “outliers” had sub-nanomolar IC₅₀s (i.e., all isolates were extremely susceptible to T111) we request that these studies be deferred. Additional clarification has been added in the revised manuscript.

4. Discrepancy between *in vitro* and *in vivo* activity. The marked discrepancy between the *in vitro* activity (IC₅₀ values in the picomolar range) and the required *in vivo* curative dose in a mouse malaria model system (50 mg/kg) raises several questions that require further investigation. Possible explanation include: poor oral bioavailability of the compound, high metabolic turn-over in mice, poor responsiveness of *P. yoelii* due to altered uptake or metabolic stability of the compound. To address these critical issues the following experiments should be performed: i) PK in mice including determination of whether parasite killing is C_{max} or AUC driven; ii) metabolism of the compound using liver microsomes from mice and humans; iii) uptake and efflux kinetics of the compound by *P. falciparum* infected human erythrocytes and *P. yoelii* infected mice erythrocytes (Is uptake mediated by NPP?); iii) target validation in *P. yoelii* with regard to *cyt b* and *pfhdhdh*; and iv stability of the compound in short term *P. falciparum* and *P. yoelii* cultures.

Response: Per suggestion, we studied *in vitro* metabolic stability of T111 using liver microsomes from human, mouse, and rhesus as well as *in vivo* PK studies in both male and female mice. The results are included in the revised manuscript. We agree that the discrepancy between the *in vitro* activity and *in vivo* curative dose may be the result of poor oral bioavailability and we have started the development of prodrugs of T111 to enhance oral efficacy and PK profiles. This approach was recently validated by early investigation of prodrugs using a T111 analog (T226), with marked enhancement in both *in vivo* oral antimalarial efficacy and PK profiles (e.g., C_{max} and AUC) in both plasma and liver (the findings were reported in our recent publication PMID: 40666482). In addition, we tested T111 in mice infected with *P. yoelii* parasites carrying *cyt b* mutations at either the Q₀ or Q₁ site, and the result is consistent with our *in vitro* observations. The findings are included in the revised manuscript. It is important to emphasize that the significance of our work is the novelty and overall profile of the acridone antimalarial chemotype. Given the paucity of anti-relapsing compounds and narrow safety profiles with tafenoquine, our acridone antimalarial offers potential for a novel radical cure (killing relapsing liver stage parasites) drug. T111 is presented as a prototype. Recent prodrug development and optimization, beyond the scope of this manuscript, should result in substantive improvements in PK profiles and oral efficacy. Other in-depth studies suggested by the

reviewer (e.g., uptake and efflux kinetics of the compound by *P. falciparum* infected human erythrocytes and *P. yoelii* infected murine erythrocytes) will be fully explored upon the selection of a late-stage preclinical candidate. In response to the reviewer's concerns, we have added to the Discussion in depth consideration of the limitations of T111 and the likelihood that these limitations can be surmounted by a prodrug strategy that is now underway.

5. Molecular targets of the compound. It is stated that *cyt b* and possibly *pfhdh* are targeted by the compound. It is unclear whether this statement is based on whole genome sequencing of resistant lines or on targeted sequencing of just the two loci. If the latter is the case, than other important resistance conferring mutations may have been missed. It is therefore recommended to determine the whole genome of some resistant line, possible those that display different levels of responsiveness to evaluate evolution of resistance. It is further recommended to evaluate the contribution of the two resistance associated genes independently using e.g., CRISPR Cas9 genome editing technology.

Response: Per the Reviewer's recommendation, we conducted whole genome sequence on 6 drug pressured clones and now report genome-wide mutations in Supplemental Table S3. The T111 mutants were generated by Dr. Roland Cooper's team, and sadly he passed away in 2024. Recently, we were able to locate the T111-selected mutant lines that were generated by his team and to submit them for whole genome sequencing. We focused WGS on six of the T111 mutant lines with 3 or 4 *pfcytb* mutations. All mutations previously identified by targeted Sanger sequencing of *pfcytb* and *pfhdh* were confirmed by WGS. Importantly, no additional recurrent mutations outside of *pfcytb* or *pfhdh* were identified that could be clearly linked to drug resistance. The remaining variants detected appear to be sporadic, randomly distributed across the genome, and likely reflect background mutations accumulated during prolonged continuous in vitro culture (over a year) rather than drug-selected resistance mechanisms. We agree that using CRISPR Cas9 genome editing technology to further investigate the T111 mutant lines would be exciting, but genetic modification of the *cytb* is not possible because it is located on the mitochondrial genome in *P. falciparum* and, to date, modification of this genome has not been reported (clarification on this has been provided in the revised manuscript). Indeed, we already have a good understanding of the mechanism of action of T111; the data presented (and new structural data; see response #6 below) suggest that the *cytochrome b* Q_o site is the primary target, as Q_o site mutations are found in all of the resistant clones. Further mechanistic studies are planned, but we feel that these will best be done after candidate selection, rather than with our current lead compound.

6. Inhibition of *cyt b* by the compound. Molecular dynamics simulations should be performed to better understand the interaction of the compound with *cyt b* at the molecular level and how mutations in the protein cause resistance. Additionally, in vitro binding studies using purified *cyt b* protein would help clarify the mechanism of inhibition and the impact of resistance-associated mutations on binding kinetics.

Response: We have developed molecular docking models of T111 and studied the selected *cyt b* mutations to predict binding of T111 to the crystal structure of *S. cerevisiae* *cyt b*,

since the sequence of *P. falciparum* cyt b is highly homologous to that of yeast cyt b. The results, showing relevant Q_o residues that were associated with T111-resistance, are now included in the revised manuscript. Furthermore, a team led by an expert structural biologist, Dr. Alexander Mühleip at University of Helsinki, recently studied binding structures of atovaquone, ELQ-300, and T111 using the cryo-EM structure of *pfbc1*. The results indicated that the binding pockets for T111 primarily reside in the Q_o region. Given this major breakthrough in the determination of the *pfbc1* cryo-EM structure to study ETC antimalarial inhibitors, a comprehensive analysis with comparison of atovaquone, ELQs, and T111 analogs will best be reported by Dr. Mühleip's group in a future manuscript focusing on high-resolution visualization of inhibitor-bound binding sites.

7. PK/PD model in humans. The development of a human PK/PD model is critical to predict clinical efficacy and inform dose selection. The authors should construct such a model to define the relationship between dosage, efficacy, and safety in humans.

Response: Our manuscript is intended to present the breakthrough discovery of a novel antimalarial chemotype. As mentioned above, it is the overall drug profile (in particular action against all parasite stages) rather than the single compound that is our focus. We offer T111 as a proof-of-concept, not as a therapeutic candidate. In response to the reviewer's concerns, we have added to the Discussion in depth consideration of the limitations of T111 and the likelihood that these limitations can be surmounted by a prodrug strategy that is now underway.

Reviewer #2:

1. A consistent number of significant figures should be used for the IC₅₀ and EC₅₀ data.

Response: The numbers in all the tables have been input with consistent number of significant figures in the revised manuscript.

2. The authors should generate in vitro ADME/PK data or discuss existing in vitro ADME/PK data for their lead compound T111 and its prodrug.

Response: The ADME/PK data for T111 has been added in the revised manuscript. The development of T111 prodrugs is underway and the findings will be reported in our future publications.

3. Why did the authors not assess if T111 also had activity against hypnozoites?

Response: We believe that these extensive studies are most appropriate after selection of a candidate compound, either the T111 prodrug or another analog, and thus that these studies are beyond the scope of the current manuscript. A prodrug of T111 is being tested for anti-relapse efficacy and synergy with tafenoquine using non-human primates infected with *P. cynomolgi*. The results, as well as extensive optimization of the acridone prodrugs, will be reported in future publications.

Reviewer #3:

Review of 'Potent Acridone Antimalarial against All Three Life Stages of Plasmodium' by Kancharla et al.

Brief summary of the most noteworthy results:

The authors of this manuscript have performed extensive experimental testing of the potential new antimalarial compound T111 using different malaria species and different stages of the life cycle (blood stage, gametocytes, mosquito stages, liver stages), using a combination of in vitro and in vivo experiments. All the malaria species in all the stages of the life cycle were eliminated by this compound, even hypnozoites in radical cure mode (the most difficult stage to treat) were susceptible. Several drug resistant *P. falciparum* strains were tested and almost all of those were still susceptible to treatment with T111. Induction of resistance was difficult and could be pinpointed to a set of mutations in CytB Q0 site.

Toxicity testing showed hardly any toxicity issues for this compound, making the compound very promising.

The experimental design is good and execution of the experiments is sound. New antimalarials are desperately needed and this compound could be beneficial for a lot of people. Hopefully this product will proceed to clinical testing and can make a difference in the combat against malaria in all its shapes and forms.

Validity

Relevant numbers of replicates have been performed for all of the experiments, the data look consistent and standard deviations are within the expected range.

Significance

The combat against malaria could benefit from the use of this compound in a clinical setting. This set of experiments is showing the potential of T111 as new antimalarial in several in vitro/vivo models. Especially interesting is the observation of activity of T111 against hypnozoites in a radical cure setting, as this would be a new scaffold replacing the much debated 8 aminoquinolines. Or it could be given as a combination drug thereby lowering the tafenoquine dose drastically.

Data and methodology

The data are presented clearly and the number of biological and technical replicates are sufficient.

It surprised me that, although many *P. falciparum* strains carrying various drug resistance mutations were tested in blood stage assays, no experiments were performed testing the efficacy on *P. vivax* blood stages (or *P. cynomolgi* as a proxy for *vivax*).

Response: We agree that assessment of blood-stage activity against *P. vivax* would be valuable. However, routine continuous in vitro culture of *P. vivax* blood stages is not possible, and ex vivo assays are limited by low throughput and availability. We therefore focused our blood-stage resistance profiling on *P. falciparum*, where well-validated panels of drug-resistant strains exist, and complemented this with studies of the liver stage of *P.*

cynomolgi, a close relative of *P. vivax* that is easier to study and the accepted surrogate for *P. vivax* relapse biology. Together, these approaches allowed us to interrogate both resistance liability and radical cure potential within practical experimental constraints. For in vivo assessment in the animals, a prodrug of T111 is being tested currently for anti-relapse efficacy and synergy with tafenoquine using non-human primates infected with *P. cynomolgi*. The results, as well as extensive optimization of the acridone prodrugs, will be reported in our future publications. Further clarification has been added in the Discussion.

I also have some remarks on the methodology of the in vivo testing of the synergy between tafenoquine/T111. The model system used (*P. yoelii* in mice for blood stage testing and *P. berghei* in mice for liver stage testing). To my knowledge tafenoquine alone is not used to treat blood stages because of the high dose needed to eliminate all gametocytes. Rather than looking at clearance of the blood stage alone, it would have been desirable to test the effect of the drug combinations on transmission. Moreover, the liver stage testing of tafenoquine and T111 in *P. berghei* infected mice does not include testing of T111 against hypnozoites, as this stage of the life cycle is not observed in this model. Although the in vitro activity against hypnozoites is promising, it is not supported by any in vivo evidence.

Response: Our prodrug approach has produced compounds with substantially enhanced PK profile and oral efficacy. A pilot safety study of T111 prodrug in rhesus monkeys suggested that the drug was well-tolerated by the animals. Validation of anti-relapse efficacy of the acridone prodrug and synergy with tafenoquine in non-human primates infected with *P. cynomolgi* is underway in collaboration with WRAIR and AFRIMS. However, these studies are not complete, and are beyond the scope of this manuscript. The results will be reported in our future publications focusing on the optimization of the prodrugs. Further clarification has been added in the Discussion.

Analytical approach

In general, the analyses are clear and straightforward. I only wonder about the Kruskal-Wallis test used for significance calculation in fig 1E, to compare oocysts of untreated vs oocysts of treated. Does not seem to me that a multiple-comparison test is needed and significance also for the lower dose would probably be shown if the statistical analysis would include a one-way ANOVA test (BUT I'm not a statistician!).

Response: A one-way ANOVA with Kruskal-Wallis multiple comparison test was used in Figure 1D (now 1F in the revised manuscript), not Figure 1E. There was no comparison between untreated oocysts vs treated oocysts in this study, as the experiment was for sexual blood stage gametocytes. In the sexual blood stage gametocytes assay, we used a Kruskal-Wallis test followed by post hoc comparisons versus the DMSO control because multiple treatment groups were analyzed simultaneously and data did not meet assumptions required for parametric testing given the small sample size. While a one-way ANOVA is less conservative, post hoc multiple-comparison correction is still required when making pairwise comparisons, and we therefore kept the nonparametric approach. In Figure 1E (now Figure 1G in the revised manuscript), the Friedman test – a test similar to ANOVA, but for non-parametric data, was used to study the inhibition of *P. falciparum* oocyst development in mosquitoes after direct membrane feeding.

Suggested improvements

This manuscript is well written, contains a huge amount of data, which are presented very clearly and without any doubt T111 is a promising compound for progression into clinical testing. The inclusion of in vivo testing of T111 as a radical cure would test the hypothesis of the activity of this compound against established hypnozoites. For this I would propose to include a set of experiments in which both prophylactic and radical cure testing of T111 (either by itself or in combination with tafenoquine) is performed on either humanized mice/*P. vivax* or rhesus monkeys/*P. cynomolgi*.

Response: As mentioned above, validation of anti-relapse efficacy of a T111 prodrug and synergy with tafenoquine using non-human primates infected with *P. cynomolgi* is underway in collaboration with WRAIR and AFRIMS. However, these studies are not complete, and are beyond the scope of this manuscript. The results will be reported in our future publications focusing on the optimization of the prodrugs. Further clarification has been added in the Discussion.

I have categorized other minor improvements or questions according to the sequence in the manuscript

Regarding blood stage testing

In Fig 1B the authors show that there is quite a lot of variation in the IC50 measured in the clinical isolates, although the IC50's are still low. Would that be an indication of the presence of parasites already carrying one or two mutations in the CytBQ0 site?

Response: Please see comment #3 to Reviewer 1 above.

Regarding the in vivo testing against *P. yoelii* blood stages (page 6 and fig 1C), I wondered how it was decided upon the treatment doses (40 and 50 mg/kg) tested in the mouse experiments. Linked to that, were there any Pk/PD tests performed beforehand using these doses? I'm asking because it looks like the difference in treatment results between the 40 and 50 mg/kg are quite striking, so I wondered if this can be explained somehow.

Response: The difference in oral efficacy in mice was only observed in blood stage testing in the single dose cure model, and we suspect that the poor aqueous solubility of T111 contributed to variability. When the compound was dosed via gavage it was not completely dissolved in the vehicle, and so provided as a suspension. To address this concern, we have added this limitation to our Discussion. Of note, T111 prodrugs provided much improved solubility and consequently enhanced PK profile and oral efficacy. These data will be disclosed in a future publication focusing on the development of T111 prodrugs.

Also, the authors state that there is no male-female difference in Pk/PD parameters, but it looks to me that the T1/2 is shorter in females (table 3), can you check if there is a statistically relevant difference in T1/2?

Response: We have now included the PK study with data for male and female mice. PK profiles of T111 were similar in male and female mice (Table 3). We also conducted a comparison of oral efficacy of T111 in the 4-day suppression rodent model with male and female mice infected with *P. yoelii*. The ED₅₀ and ED₉₀ values in male and female mice were almost identical (Figure S1). These results are included in the revised manuscript.

In the text it is written ‘long half-life’ (p17). Do the authors think 15-18hr half-life is long? If you compare this to the T_{1/2} of other antimalarials (T_{1/2} of chloroquine is 20-40 days, T_{1/2} of tafenoquine is 14 days T_{1/2} of atovaquone is 5-6 days), the T_{1/2} is not very long.

Response: The listed T_{1/2} values of other antimalarials by the reviewer are in humans. Tafenoquine has a T_{1/2} of 50 h in mouse plasma and 75 h in mouse liver (internal data from WRAIR). The T_{1/2} for chloroquine in healthy mice is about 46 h (PMID: 21646487; PMCID: PMC3147625). The T_{1/2} for atovaquone in healthy mice is about 13 h (PMID: 23292347; PMCID: PMC4344550). T111 T_{1/2} in mouse plasma ranges from 17 h to 26 h (Table 3). To clarify this point, we added to the revised manuscript that this is a relatively long half-life for an antimalarial compound in mice.

Gametocyte stages

Suggestion for the layout of the paper: It would be more logical to put the mosquito stage testing after the gametocytes if the manuscript follows the mosquito life cycle, which it did up to this point.

Response: This is a great suggestion and we have moved the mosquito life cycle section immediately following the gametocytes studies.

Liver stages

The activity of T111 is compared to the reference drugs Atovaquone and tafenoquine. So why is maduromycin in the table? The authors also state in the discussion that atovaquone is not active against hypnozoites, but the IC₅₀ of atovaquone against hypnozoites is quite low. That is contradictory.

Response: The in vitro anti-relapse activity in the radical cure mode for tafenoquine heavily depends on P450. Maduromycin was included as a positive control because it has potent activities against hypnozoites and schizonts in both prophylactic mode and radical cure mode. However, the anti-relapse efficacy of tafenoquine was validated using *P. cynomolgi* infected rhesus monkeys. Although atovaquone shows in vitro activity against hypnozoites, it does not provide a durable radical cure and is ineffective at preventing relapse in vivo. In our *P. cynomolgi* rhesus hepatocyte system, however, this weak activity (IC₅₀ values > 2 μM) occurred at concentrations that overlap with hepatocyte cytotoxicity (Table 1, Source data file), resulting in a narrow or absent therapeutic window. Furthermore, independent literature provides in vitro liver-stage atovaquone IC₅₀ values consistent with those observed by us (≈2 μM) in primary hepatocyte *P. cynomolgi* assays, PMID: 21483865). Importantly, those studies did not report a therapeutic index, and as noted in our work, apparent atovaquone hypnozoite inhibition occurs at concentrations overlapping with cytotoxicity. Since the submission of our original manuscript, additional

experiments with independent biological replicates have been performed and Table 1 has been updated. We also added an additional reference drug KDU691 (PI4K) inhibitor) with prophylactic activity against liver stage parasites, but ineffective as a radical cure agent against relapsing hypnozoites (PMID: 26926645).

Does T111 eliminate the hypnozoites from the cultures completely or do the authors still see tiny parasites after prophylactic or radical cure treatment?

Response: The high throughput in vitro hypnozoitocidal assay developed by WRAIR using *P. cynomolgi* sporozoites and non-human primate rhesus hepatocytes was conducted with immunofluorescence staining and imaging by an Operetta CLS high-content imager with high sensitivity. Following T111 treatment in prophylactic mode, no morphologically detectable hypnozoites were observed above background levels at the assay endpoint (raw counts available in the Source data file), although we cannot exclude the presence of non-viable residual structures below detection limits. Furthermore, following T111 treatment in radical cure mode, a small residual population (~10-20%) of hypnozoites remained with the highest concentrations tested, as shown in the source data file. Future studies will be conducted to determine whether the residual hypnozoite population observed under radical cure conditions remains viable using extended-duration liver stage assays.

Mosquito stages

In the tarsal contact tests, reduced oocyst numbers were seen. But do these still result in sporozoites? And if so, are the sporozoites infective?

Response: We did not assess sporozoite numbers in these experiments, so we cannot be certain whether these few oocysts produced infectious sporozoites. However, we can be certain that the vast majority of tarsally exposed mosquitoes will produce no sporozoites, given the 92.5% reduction in oocyst prevalence, and that those mosquitoes with breakthrough infections will produce few sporozoites, given the 97.6% reduction in oocyst intensity. Oocyst counts following the standard membrane feeding assay (SMFA) are considered the gold standard for transmission blocking studies (PMID: 24042109, PMID: 25890243, PMID: 29247222) and are positively correlated with subsequent sporozoite numbers (PMID: 31060594, PMID: 38272943, PMID:38517746). Moreover, it has been shown that sporozoite transmission is rare at inoculums below ~10,000 sporozoites (PMID: 32453765). Given the extremely low prevalence and intensity of oocysts in these experiments, we expected that these three infected mosquitoes would likely not reach this threshold of sporozoites, and we therefore chose not to measure sporozoites in this experiment. We do not expect that pre-infection T111 tarsal contact would have activity against sporozoites directly because it has an estimated half-life of ~18 hours. However, it is possible that development of the few breakthrough oocysts we observed and their subsequent sporozoite infectivity may be impaired in some way by the lingering impact of T111 exposure during initial development. Due to the low infection prevalence and intensity, we would be unable to isolate sufficient sporozoites to perform robust liver stage infections to assess this possibility, and we therefore chose not to discuss this in the text. Further, we think that extensive additional experiments to precisely characterize

quantitative impacts on transmission will be most appropriate with a T111 prodrug after candidate selection, as discussed above, rather than at this early stage in drug development.

Was the transmission blocking effect tested only by adding the compound to infected blood? I think these kinds of tests are suitable as a first screening, but feeding on infected/treated animals would be additionally informative with regards to transmission testing.

Response: The transmission blocking effects of T111 were assessed using three different assays: 1) sugar bait feeding assay where drugs were mixed into a sucrose-water solution and fed to infected mosquitos (reported by our collaborator Dr. Dean Goodman in previous publication PMID: 40449870); 2) direct membrane feeding assay (SMFA) that simulates a blood meal taken by the mosquitos from a treated human (presented in this manuscript), and 3) the tarsal contact assay that mimics uptake of the drugs via mosquito's legs after landing on antimalarial infused bed nets, even with short exposure time (presented in this manuscript). SMFA is the gold standard for testing transmission blocking in a preclinical study. This assay specifically tests the impact of a drug on the ability of functional gametocytes to infect mosquitoes and avoids confounding effects of the drug on asexual growth and gametocyte development prior to infection. Specific impacts on asexual and gametocyte development are demonstrated in other experiments (Figure 1). Treating animals prior to transmission is possible, but the impacts of the drug on asexual growth and gametocyte development prior to infection confound the analysis of stage specific effects, so these trials are not usually done at this stage.

Fig 1E what is the oocyst load of the untreated control?

Response: The oocyst loads of the untreated control vary between biological repeats, so the data have been normalized to better compare the impact of the drug across multiple trials. Raw counts of oocyst load can be found in the source data file for Figure 1G in the revised manuscript.

Synergy experiments

The synergy between T111 and tafenoquine is obvious. Inhibition of the Q_0 by T111 increases the parasite's sensitivity for tafenoquine. Does this also help to identify the mode of action of tafenoquine?

Response: This is a great question. The mechanism of action of tafenoquine is not fully understood, but its radical curative activity is likely due to its metabolites, which generate reactive oxygen species that may damage the parasite's cellular components. Studies suggest that tafenoquine may also cause mitochondrial dysfunction and an increase in intracellular calcium levels within the parasite. One possible explanation of synergy is that the mitochondrial disruption caused by tafenoquine leads to vulnerability to other drugs targeting mitochondrial ETC. But, to our knowledge, other ETC inhibitors such as atovaquone and ELQs are not synergistic with tafenoquine. Our studies differentiating between T111 and tafenoquine in their effects on G6PD deficient erythrocytes suggest that T111 does not produce redox cycling. Dr. Dennis Shanks, who conducted the first field trial of tafenoquine, is investigating the effect of T111 and its prodrugs on the metabolism and

redox cycling of tafenoquine. The results will best be reported by Dr. Shanks' group in a future manuscript including a large panel of standard antimalarials.

Fig 2C Are the T111 alone curves not included?

Response: This figure is intended to focus on the leftward shift (potency increase) achieved by the addition of T111, compared to tafenoquine alone. The inclusion of T111 alone curve would interfere visually. However, the dose-response curves for T111 alone are shown in Figure S1, which was added in the revised manuscript. Importantly, the doses of T111 used in this combination study were below therapeutic level (ED_{50}/ED_{90} of T111 = 0.27/0.39 mg/kg/d).

I found a typo in page 29

Header: Drug subspeciality testing for cross-resistance profile.

I assume the authors mean susceptibility here

Response: Thanks! The typo has been fixed in the revised manuscript.